# Covalent transfer of chemical gradients onto a graphenic surface with 2D and 3D control

Yuanzhi Xia[1,7], Semih Sevim[2,7], João Pedro Vale ®[3,4,7], Johannes Seibel ®[1], David Rodríguez-San-Miguel ®[5], Donghoon Kim[2], Salvador Pané ®[2], Tiago Sotto Mayor ®[3,4] ✉, Steven De Feyter ®[1] ✉ & Josep Puigmartí-Luis[5,6] ✉

Control over the functionalization of graphenic materials is key to enable their full application in electronic and optical technologies. Covalent functionalization strategies have been proposed as an approach to tailor the interfaces' structure and properties. However, to date, none of the proposed methods allow for a covalent functionalization with control over the grafting density, layer thickness and/or morphology, which are key aspects for fine-tuning the processability and performance of graphenic materials. Here, we show that the no-slip boundary condition at the walls of a continuous flow microfluidic device offers a way to generate controlled chemical gradients onto a graphenic material with 2D and 3D control, a possibility that will allow the sophisticated functionalization of these technologically-relevant materials.

Continuous-flow microfluidic devices have been long used for materials synthesis[1–3]. The spatiotemporal control of reagent flows inside a microfluidic channel allows for the generation of a controlled reaction-diffusion (RD) environment[4], where compositional gradients in solution can be precisely controlled. In a pioneering contribution, Whitesides and co-workers demonstrated that the RD condition achieved inside a continuous flow microfluidic device could, for example, be used to pattern Ag films onto a glass substrate[1]. Downard et al. also employed a continuous flow microfluidic device to pattern in-situ formed aryldiazonium salts on pyrolyzed photoresist film (PPF) under controlled RD conditions[5]. Although these two studies demonstrated that the width of the films generated could be tailored by changing the flow rates, the functionalization of the surfaces investigated in both cases occurred only in a narrow region relative to the width of the microfluidic channel. Furthermore, the studies did not describe any attempt to take advantage of the compositional gradients to do controlled functionalization of large surface areas, which is important to boost developments in sensor technologies, electronic devices, and advanced catalysts, to name a few. To a great extent, the control over compositional gradients achieved in solution and inside continuous flow microfluidic devices has been mainly used to understand self-assembly processes involving crystal growth[6–9], to isolate out-of-equilibrium assemblies[10,11], to unveil materials with new properties and functions[12,13], and to induce chirality in supramolecular aggregates[14].

Interestingly, while the no-slip boundary condition prevailing at the walls of continuous-flow microfluidic devices is often associated with precipitation events that may limit the device's long term operation and performance[6], it can also offer a powerful way to control surface reactions, an aspect which has been overlooked in prior studies[15]. Herein we demonstrate that the no-slip boundary condition can be leveraged to enable a spatiotemporal functionalization of millimeter-size reactive surfaces in 2D and 3D, with a nanometric thickness control. Our results demonstrate that our microfluidic approach offers a reliable high-throughput methodology to study the effect of multiple compositional libraries onto technologically-relevant reactive surfaces in a reliable, cost- and time-efficient manner.

[1]Department of Chemistry, Division of Molecular Imaging and Photonics, KU Leuven, Leuven, Belgium. [2]Multi-Scale Robotics Lab, Institute of Robotics and Intelligent Systems, ETH Zurich, Zurich, Switzerland. [3]Transport Phenomena Research Centre (CEFT), Engineering Faculty of Porto University, Porto, Portugal. [4]Associate Laboratory in Chemical Engineering (ALiCE), Engineering Faculty of Porto University, Porto, Portugal. [5]Departament de Ciència dels Materials i Química Física, Institut de Química Teòrica i Computacional, University of Barcelona (UB), Barcelona, Spain. [6]Institució Catalana de Recerca i Estudis Avançats (ICREA), Barcelona, Spain. [7]These authors contributed equally: Yuanzhi Xia, Semih Sevim, João Pedro Vale. ✉e-mail: tiago.sottomayor@fe.up.pt; steven.defeyter@kuleuven.be; josep.puigmarti@ub.edu

**Article** https://doi.org/10.1038/s41467-022-34684-w

## Results and discussion

### Controlled covalent functionalization with microfluidics

To control the compositional chemical gradients inside microfluidic devices and enable their transfer onto reactive substrates, we designed a continuous-flow microfluidic device composed of two machined layers positioned one on top of the other (Fig. 1a). The top layer consists of a Y-shaped microchannel and connection ports for two inlets and one outlet, whereas the bottom layer serves as a support to mechanically clamp the Y-shaped microchannel onto reactive substrates (see Design and fabrication of microfluidic cell in Methods, for further details). The Y-shaped microchannel is the simplest configuration enabling the generation of precise compositional gradients between two reactants injected through the two inlet ports – a configuration where gradients can be perfectly controlled due to the laminar nature of the flow and the associated mixing by molecular diffusion[16]. Importantly, by changing the flow rate of the reactants and the total flow rate, one can control the RD area (Fig. 1b), in particular the space and time over which the two reactive streams are allowed to diffuse into each other throughout the main microfluidic channel, until reaching the outlet port. This offers precise control over the spatio-temporal compositional gradient generated between the two reactive streams flowing inside the main microfluidic channel[17].

To confirm that our microfluidic device induces a laminar flow condition, we initially injected two aqueous solutions of a blue and red food dyes into the device's inlets, and observed the two colored streams flowing side-by-side along the main microfluidic channel, without any signs of leakage or local disturbances (Fig. 1c and Supplementary Fig. 1). For the proof-of-concept, we chose highly oriented pyrolytic graphite (HOPG) as the reactive surface. HOPG is a layered model system that is often used in scanning probe microscopy studies, and whose large atomically flat terraces and low defect density makes

it a suitable substrate for our investigations. The top layer of the microfluidic device, which is made of polyether ether ketone (PEEK) or polymethyl methacrylate (PMMA), contains a sealing groove around the edges of the Y-shaped microchannel that, once pressed onto the HOPG, allows to enclose and completely seal the whole microfluidic network (Supplementary Fig. 2). Even though HOPG is considered a chemically inert surface, we have recently reported that aryl radicals can be efficiently grafted onto HOPG when a diazonium salt is reduced with potassium iodide (KI, Fig. 1d)[18]. Additionally, we also showed that the nitrogen gas generated during this reaction can induce the formation of corrals on the HOPG substrate[18], which prevent a precise control over the covalent chemical derivatization of HOPG in 2D and 3D. For these reasons, the HOPG and this reaction protocol is an excellent model system to illustrate the effect of the no-slip boundary condition, on a controlled functionalization of a graphenic surface in 2D and 3D. It should be noted that the size and location of the RD area determines the location where the aryl radicals are generated, and therefore, where the grafting of the HOPG will take place, when the aryl radicals pass close to it. Moreover, a dendritic growth is also possible when the aryl radicals generated are allowed to react with molecules previously grafted onto the HOPG substrate, which provides control over the thickness of the functionalized layer[19–22] (Fig. 1e).

In a typical microfluidic experiment, aqueous solutions of *para*-nitrobenzenediazonium-tetrafluoroborate (NBD, 10 mM) and potassium iodide (KI, 10 mM) were simultaneously injected from the two inlet ports (Fig. 1a), at a constant flow rate of $50\,\mu L\,min^{-1}$ for 5 min (see Microfluidic experimental procedure in Methods). Under these conditions, the two reactant-laden streams flow side-by-side along the main microfluidic channel, with mixing at the interface occurring by molecular diffusion only[23–25]. Since both reactants have different diffusion coefficients ($8.14 \times 10^{-10}\,m^2\,s^{-1}$ for NBD determined by NMR

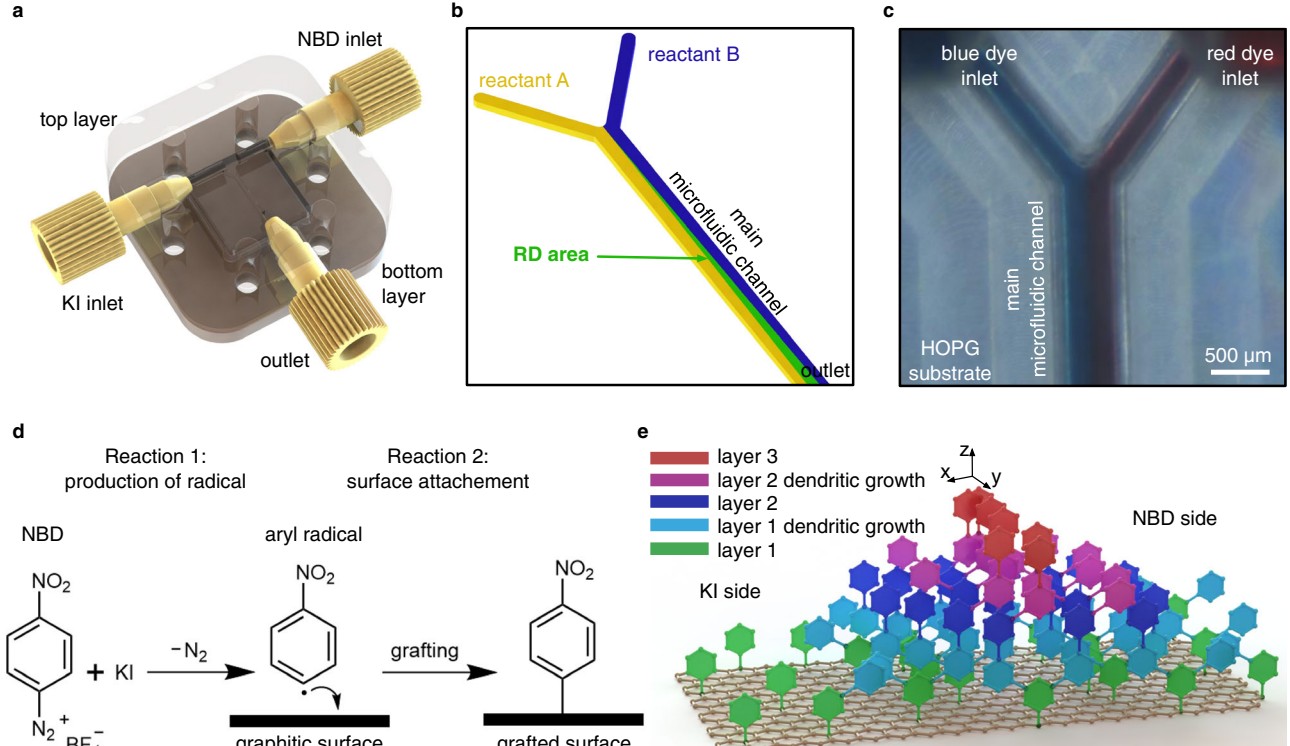

**Fig. 1 | Controlled covalent functionalization of highly ordered pyrolytic graphite (HOPG) under microfluidic conditions. a** Schematic drawing of the assembled microfluidic device including a Y-shaped microchannel featuring two inlets and one outlet connection ports. **b** Illustration of the main microfluidic channel showing the RD area generated inside a continuous-flow microfluidic device. **c** Micrograph of the assembled microfluidic device showing the laminar condition achieved with two food dyes over the HOPG substrate. No leakages or local flow disturbances were observed. Scale bar: 500 μm. **d** Schematic reactions of NBD with KI on a graphitic surface. **e** Illustration showing the dendric growth that can occur with the reactions presented in **d**.

**Nature Communications** | (2022)13:7006 2

spectroscopy, see Supplementary Note 1, and $2 \times 10^{-9}$ $m^2 s^{-1}$ for KI[26]), the concentration gradients forming along the width (x-axis) of the main microfluidic channel should be slightly asymmetric, with the concentration profile of aryl radicals slightly shifted towards the NBD side of the microfluidic channel due to the faster diffusion of KI.

## Mass transport within the microfluidic device

To investigate the formation of concentration gradients inside our microfluidic device and over the HOPG substrate, we numerically simulated the transport processes occurring in the microfluidic device, mimicking all the experimental conditions (i.e. a flow rate of $50 \mu L$ $min^{-1}$ and a concentration of 10 mM for each reactant-laden stream) and the device geometrical features, after validating the modelling approach and its predictions (Supplementary Figs. 3–5). To consider the effect of the formation and consumption of aryl radicals within the main microfluidic channel, four reactions were considered in the simulations, i.e. (i) the production of the aryl radical; (ii) the consumption of aryl radical by the surface attachment (grafting); (iii) the formation of iodine ($I_2$); and (iv) the deactivation of the aryl radical with $I_2$ (see Numerical simulations in Methods, and Supplementary Notes 2–5)[27].

The concentration of aryl radical was numerically predicted in three-dimensional (3D) simulations (Fig. 2a), and particular focus was put on the concentration profiles near the HOPG substrate, to enable studying the favorable effect of the no-slip boundary condition over the grafting of the aryl radical onto the HOPG substrate inside our microfluidic device (Supplementary Notes 4 and 5).

The numerical results confirmed the faster diffusion of KI into the coflowing NBD-laden stream because of the higher diffusion coefficient, and the formation of an asymmetric concentration map of aryl radicals that is slightly shifted towards the NBD side (Fig. 2b). Note that Fig. 2b, c display more diffusion near the top and bottom walls of the microfluidic channel because of the higher residence time caused by the no-slip boundary conditions prevailing at the walls of the main microfluidic channel. (Supplementary Note 4 and Supplementary Figs. 6–9 for details on the effect of the no-slip boundary conditions). Consequently, the reactant molecules flowing near the top and bottom of the microfluidic channel have more time to diffuse and react to form the aryl radicals. A closer inspection of Fig. 2b, c also shows a concentration of aryl radicals near the HOPG substrate that is one-to-two orders of magnitude smaller than that near the top wall, because of its concomitant consumption in the reaction with the HOPG substrate. Yet, there is a non-monotonic variation in the concentration of radicals near the HOPG substrate (Fig. 2d) as the reactants move along the channel length (y-axis) and participate in the various reactions mentioned above (Supplementary Notes 2–5, Supplementary Table 1 and Supplementary Figs. 10–13 for details on the reaction mechanisms, the reaction constants and their effect over the aryl radical concentration inside the microfluidic device). The concentration of radicals increases initially from $1.5 \mu M$ at the start of the main microfluidic channel where the two reactant-laden streams meet and start diffusing ($y = 0 \mu m$, before zone 1), to ~$3 \mu M$ near the middle length of the main microfluidic channel ($y = 2000 \mu m$, before zone 2; Fig. 2d). This is a direct consequence of its continuous formation along the main microfluidic channel length (Supplementary Fig. 14), due to the mixing of NBD and KI by diffusion (Fig. 2e, Supplementary Fig. 5b). From that position onward ($y > 2000 \mu m$), the continuous formation of $I_2$ along the reactor length and the associated deactivation of the aryl radicals (Supplementary Notes 3–5) becomes more relevant, and the concentration of radicals starts decreasing from ~$3 \mu M$ ($y = 2000 \mu m$, before zone 2) to $2.5 \mu M$ at the outlet (zone 3; Fig. 2d). This happens because, in that region, the number of aryl radicals undergoing deactivation is higher than those being produced plus those diffusing from the bulk to the substrate (Supplementary Fig. 14). The variation in concentration of aryl radicals along the main microfluidic channel length can also be observed when comparing the profiles of aryl radical concentration along its width (x-axis) for various positions along its length (zone 1, 2 and 3; Fig. 2f; see the zone color codes in Fig. 2a, d). These profiles get wider from zone 1 to zone 3, due to the increasing diffusion and reaction times. Furthermore, their maximum concentration increases from zone 1 to zone 2, and decreases from zone 2 to zone 3, due to the interplay between production and deactivation of aryl radicals (Supplementary Note 5 and Supplementary Fig. 14).

Importantly, the above numerical results indicate that the aryl radicals generated at the RD area can cover at least $300 \mu m$ of channel width at zone 2 (Fig. 2f), and the entire channel width (i.e. $400 \mu m$) at zone 3 (Supplementary Fig. 15). Additionally, it is important to highlight that, with this microfluidic approach, we can control not only the position and extension of the RD area near the HOPG substrate, but also the time the aryl radicals generated in the RD area are allowed to react with the substrate (as time varies inversely with the chosen total flow rate; Supplementary Figs. 8–9). The important implication of this is that, by controlling the position, the extension as well as the time of the RD process along the main microfluidic channel and near the HOPG substrate, one can control its functionalization in 2D and 3D. Conversely, at the mid-height of the main microfluidic channel ($z = 60 \mu m$), the much higher flow velocities and associated lower residence times (Supplementary Fig. 7 vs Supplementary Fig. 8) imply much less diffusion at the interface between the two reactant-laden streams (Supplementary Fig. 5b) and, thus, a much narrower RD area than that observed near the HOPG substrate (Fig. 2c, g). If these higher velocities would occur near the walls (i.e. in a hypothetical case in which a free-slip boundary condition would exist at the HOPG substrate, instead of the no-slip boundary condition; Supplementary Figs. 16–17), they would prevent the functionalization of large areas of the substrate because of the much smaller diffusion width (Supplementary Fig. 16f–g vs Fig. 2f–g). Thus, this confirms that the no-slip boundary condition prevailing at the walls of a continuous-flow microfluidic device is key to control the solution-based dynamic gradients near substrates, which, in turn, enable the spatiotemporal functionalization of technologically-relevant reactive surfaces in 2D and 3D.

## Spatially controlled covalent functionalization

To experimentally determine the effect of the no-slip boundary condition during the microfluidic grafting of a solution-based dynamic gradient onto HOPG, the morphology and thickness of different HOPG grafted samples were characterized using atomic force microscopy (AFM) (see Methods and Supplementary Figs. 18–19). Note that the mark generated on the HOPG substrate by the sealing grooves of the microfluidic device enabled the accurate localization of a specific position within the main microfluidic channel (Supplementary Fig. 2e), thus allowing a position-dependent characterization of the grafting process (see Methods). As shown in Fig. 3a, the representative AFM images acquired at different positions along the main microfluidic channel differ significantly depending on the location where the measurement is performed. On the left side of the main microfluidic channel, where the KI solution is flowing, the layer thickness is ~1.0 nm (approx. the height of a single aryl moiety) around zone 1 and zone 3 (red and blue graphs in Fig. 3b, respectively). At these two channel positions, the layer thicknesses are 50% larger at the channel center and on the right side of the main microfluidic channel where NBD-laden stream is flowing (1.5 nm; Fig. 3b). At zone 2, the layer thickness is smaller on the KI side (1.2 nm), larger at the center (2.0 nm) and again smaller on the NBD side (1.5 nm; black graph in Fig. 3b). This pattern of the grafting layer thickness along the x-axis (Fig. 3b) is fully consistent with the simulated aryl radical concentration profile along the same x-axis (Fig. 2f). Indeed, the asymmetric bell-shaped profile, which is slightly shifted towards the NBD side, implies the lowest aryl radical concentrations on the KI side, the largest concentrations at the center

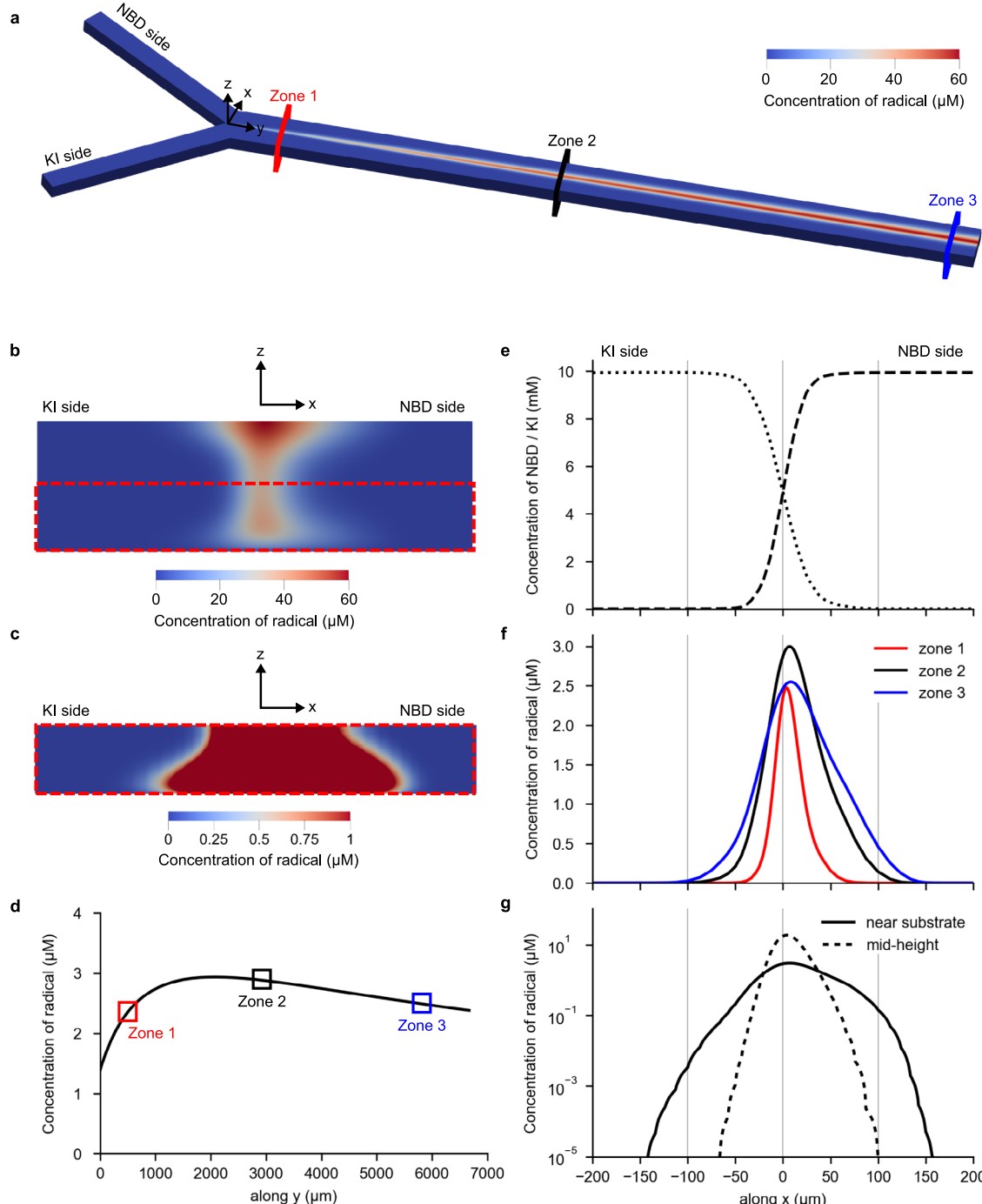

**Fig. 2 | Numerical results of the mass transport in the microfluidic device.**
**a** Concentration map of the aryl radical concentrations in the entire main microfluidic channel. **b** Concentration map of aryl radical at the channel mid-length (i.e. xz-plane, zone 2). **c** Concentration map of aryl radical at the channel mid-length (i.e. xz-plane, zone 2), considering a very narrow concentration range to show the aryl radical concentration close to the HOPG substrate. **d** Concentration of aryl radical at the HOPG substrate along the main microfluidic channel length (along y-axis, x=0). **e** Concentration profiles of the reactants (KI and NBD) along the width of the channel (along x-axis, zone 2) near the HOPG substrate. **f** Concentration profiles of aryl radical near the HOPG substrate for different positions along the channel length (zone 1 [near inlet], zone 2 [mid-length] and zone 3 [near outlet]). **g** Concentration profiles (in log scale) of aryl radical near the HOPG substrate and at the mid-height of the channel (zone 2; the curves are not smooth due to the logarithmic scale used in the vertical y-axis). Representative coordinate systems and color codes for different channel zones are given in **a**. The concentration data were calculated based on the equations of aryl radical production, deactivation and grafting, described in Numerical simulations in Methods, considering $k_1 = 10^{-3}\,\text{m s}^{-1}$ and $k_2 = 2 \times 10^5\,\text{M}^{-1}\,\text{s}^{-1}$.

and slightly lower concentrations on the NBD side. Furthermore, along the y-axis, the layer thickness increases from zone 1 to zone 2 and decreases from zone 2 to zone 3 (Fig. 3b), in full alignment with the predicted change in aryl radical concentration (Fig. 2d, f and Supplementary Note 5).

Remarkably, the solution-based dynamic gradients generated near the HOPG allow for not only controlling the layer thickness but also modulating the size, shape and number of corrals (i.e. the non-grafted empty regions) with spatial control. On the KI side of the main microfluidic channel, the corrals have irregular shapes that cover

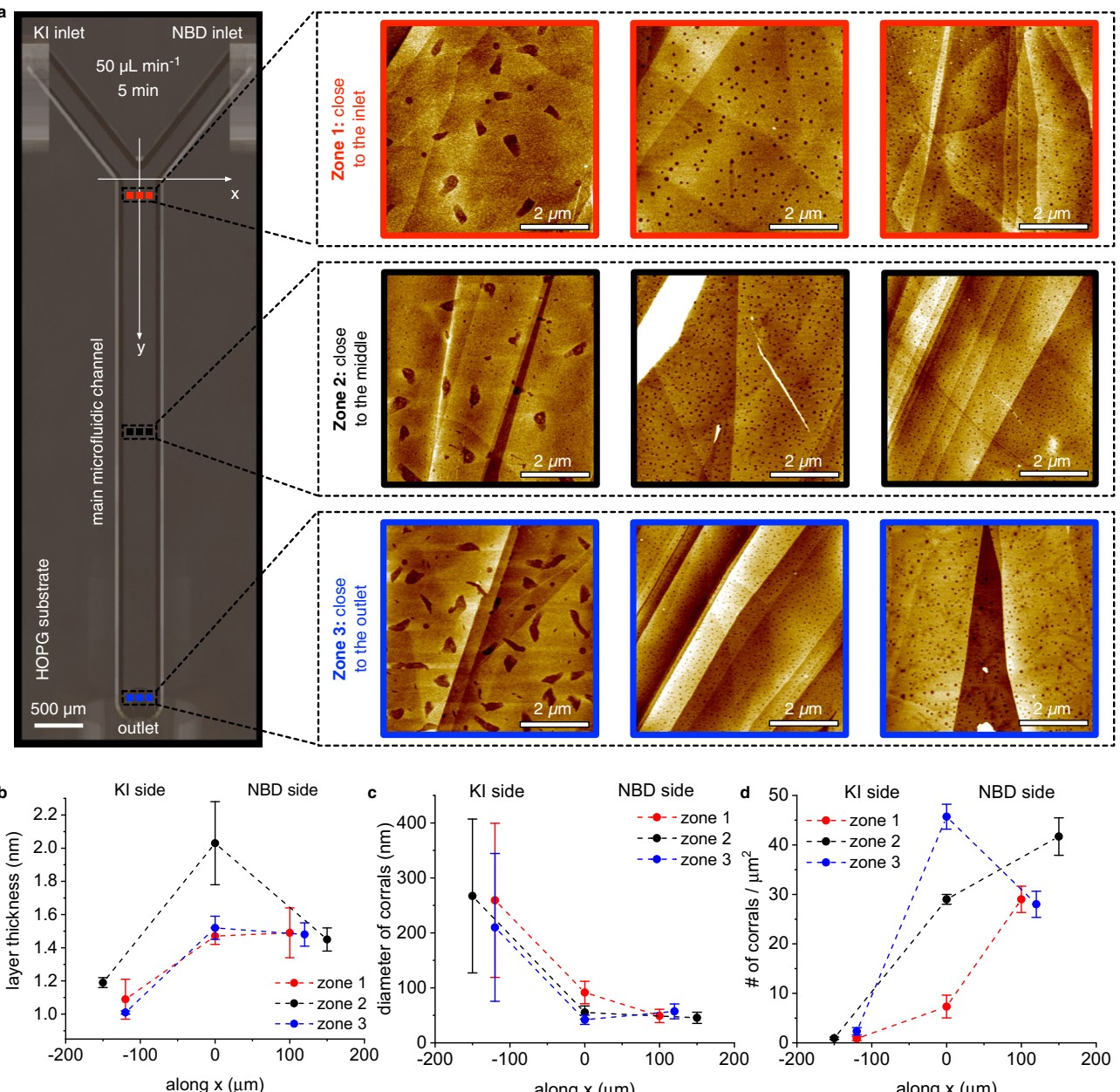

**Fig. 3 | AFM characterization of the functionalized HOPG. a** Top view of the microchannel showing the inlet and outlet ports as well as the locations from where the AFM images where acquired. The AFM images are color coded with respect to the locations in three different zones along the main microfluidic channel (red, black and blue strokes respectively for zone 1, zone 2 and zone 3). **b**–**d** Graphs showing the layer thickness, diameter of corrals and the number of corrals obtained in each zone and along the x-axis. Data points for the three representative locations along the x-axis are connected with color-coded dashed lines (red, black and blue). Error bars represent the standard deviation of at least three different measurements. The experiments were performed with a flow rate of 50 μL min⁻¹ for 5 min. Scale bars: 2 μm.

regions of ~250 nm in diameter, whereas at the center and on the NBD side of the main microfluidic channel, the corrals are circular with diameters of ~50 nm along the x-axis (Fig. 3c). As the formation of these corrals has been attributed to the generation of nitrogen gas during the reduction of NBD[28–31], the differences in their distribution and size can be attributed to the different reaction conditions achieved within our microfluidic device, where the local concentrations of NBD, KI and aryl radicals near the HOPG substrate are clearly better controlled than with the commonly used bulk methods (see Supplementary Fig. 20a). Indeed, the size of corrals is decreasing along the width of the main microfluidic channel (x-axis) towards NBD side (Fig. 3c)– i.e. along the course where the concentration of aryl radicals is higher (comparing the KI versus NBD sides in Fig. 2f). Additionally, the density

of corrals also varies drastically along x-axis from the KI side towards the NBD side, in particular at zone 2 where the number of corrals grows from less than 1 corrals/μm² at KI side up to 40 corrals/μm² at NBD side (Fig. 3d). Clearly the smaller number of aryl radicals available on the KI side yields larger corrals with a lower density (Fig. 3a), whereas the higher aryl radical concentration at the center of the main microfluidic channel and towards the NBD side, favors the grafting of the aryl radicals onto the HOPG in more uniform grafted layers featuring smaller corrals (Fig. 3d). In sharp contrast, when the same conditions are used with conventional bulk methods, irregularly shaped corrals distributed randomly onto the HOPG substrate are observed (Supplementary Fig. 20a). This result can be explained by the lack of control over the mass transport of aryl radicals onto the HOPG substrate in

time and space during the grafting process. To further highlight the efficacy of our microfluidic approach, we connected a second microfluidic device to the outlet port of the first microfluidic device (Supplementary Fig. 20b) and repeated the experiment. We observed that, while the first microfluidic device enables the spatiotemporal control of the grafting of aryl moieties throughout the main microfluidic channel, such control is lost in the second microfluidic device, with the HOPG located in the main microfluidic channel of the latter device showing a topography similar to that of the bulk sample (Supplementary Fig. 20c, d). This is not surprising, as the attachment of the ports connecting the two devices unavoidably induces flow disturbances that promote local mixing and destruction of the precise gradients achieved in the first device. Importantly, these data highlight the key importance of controlling the RD area near the HOPG substrate to be able to transfer a solution-based dynamic gradient that results from the effect of the no-slip boundary condition prevailing at the reactive substrate.

### Temporal evolution of the functionalization layer

While the previous experiments demonstrated the formation of a chemical gradient onto the HOPG (along the x- and y-axis), a controlled chemical grafting in 3D remained still a challenge with the parameters employed in our microfluidic experiments. As indicated above, in a typical microfluidic experiment we fixed the flow rate of NBD and KI at 50 $\mu$L min$^{-1}$ and the reaction time of the aryl radicals generated near the HOPG under the controlled RD conditions, at 5 min. To further investigate the effect of reaction time, we performed time dependent microfluidic-based HOPG grafting experiments considering different flow times, varying from 1 minute to 60 min, for constant initial reactants concentration and flow rates. We focused our analysis on a fixed position in the main microfluidic channel, to ease the comparison of results obtained with the different samples. To avoid errors and gain significant detailed information, we acquired the AFM images at the center of zone 2 for each sample (Fig. 4a), a location where the RD area will be sufficiently developed. Note that, for a fixed position in the channel (in this case, zone 2) and fixed initial reactants concentration and flow rates, the concentrations of NBD, KI and aryl radicals will be similar between different experiments, because of the control of the RD area offered by the microfluidic approach.

As shown in Fig. 4b, the thickness of the grafted layer increases from about 1.5 nm to 2.3 nm when the flow time increases from 1 to 60 min. This change in the thickness of the grafted layer can be explained by the increasing total number of aryl radicals passing over the HOPG substrate at this specific location. Importantly, the increase in the grafted layer thickness is not linear, as the thickness increases by 0.5 nm when the flow time is changed from 1 to 5 min, but only by 0.3 nm when the flow time is increased from 5 to 60 min. These changes in the grafted layer thickness with time show that the functionalization of the HOPG with the first layer is a relatively fast process (less than 1 min) and it is subsequently followed by a significantly slower dendritic multilayer growth (Fig. 1e).

The results presented are consistent with data obtained from Raman spectroscopy, which was used to confirm the covalent functionalization of the HOPG substrates. The sp$^3$ defects in the sp$^2$ graphite lattice during the covalent derivatization of the HOPG with the aryl moieties lead to the appearance of a characteristic $D$-band in the Raman spectrum, whereby the ratio between the appearing $D$-band and the $G$-band (I($D$)/I($G$)) ratio indicates the degree of the covalent functionalization[32] (see Methods). Raman spectra taken at different positions along the main microfluidic channel show very similar I($D$)/I($G$) ratios (*ca.* 0.010 $\pm$ 0.001), indicating that no clear differences in the number of molecules grafted onto the HOPG substrate are evident (Supplementary Fig. 21). Accordingly, these results demonstrate that the reason for the different morphologies and layer thicknesses observed is the result of the slow dendritic multilayer growth that

follows the fast initial covalent derivatization of the HOPG substrate, a process that it is well known for electrochemically grafted NBD[19,28,33]. Clearly, with these experiments, we proved that even though the reaction of aryl radicals with a previously generated aryl grafted layer is slower than the direct reaction of the aryl radicals with the HOPG, a controlled 3D functionalization is feasible provided we allow enough time for the reaction to occur under a RD condition. Note that the no-slip boundary condition achieved with the microfluidic device allows for the growth of a multiple layer due to the continuous generation of aryl radicals in the RD area near the HOPG substrate, a continuous generation that ensures access of these radicals to the substrate—in controlled conditions including concentration gradients that affect the rate of mass transport. This is in distinctly different to what happens with, for example, the commonly used drop-casting method, where such continuous access of radicals to the substrates cannot be ensured. Notably, we demonstrated that, with our microfluidic approach, an aryl grafted layer of 3 nm could be easily achieved by increasing the flow time to 120 min (Fig. 4d–g). Conversely, the number of corrals does not change significantly with the flow time as the concentration of aryl radicals is constant at this specific position. However, their average diameter decreases from about 60 nm to 40 nm when the flow time increases from 1 to 60 min, indicating that the corrals form very fast and their size reduces with time due to the covalent functionalization with more aryl radicals, as well as with the dendritic growth at the edges of the corrals (Fig. 4c).

### Effects of the flow rate on the functionalization layer

Although the above time-dependent experiments provided a clear evidence of the precise and controlled layer growth (i.e. the grafted layer thickness increases with increasing the flow time), to achieve the formation of a monolayer at the investigated location (i.e. at the middle of the main microfluidic channel) by only modulating the duration of the experiment would be unattainable and unrealistic. For instance, the time required for disassembling the microfluidic device and rinsing the HOPG substrate would not be compatible with performing experiments that last less than 1 minute, where the thickness of the grafted layer already reaches around 1.5 nm (see first data point in Fig. 4b). Instead, to change the time during which the reactants were allowed to react with the HOPG substrate to form a monolayer, we simply changed the flow rates of the reactants during the microfluidic experiment. We varied the flow rates from 50 $\mu$L min$^{-1}$ to 100 $\mu$L min$^{-1}$ and 200 $\mu$L min$^{-1}$, for a fixed flow time of 1 min (Fig. 5a and Supplementary Fig. 22). Interestingly, increasing the flow rate from 50 $\mu$L min$^{-1}$ to 200 $\mu$L min$^{-1}$ decreased the layer thickness from 1.5 nm to 1.2 nm (Fig. 5b). This indicates that the residence time of the aryl radicals being generated in the RD area close to the HOPG substrate was playing an important role in the thickness of the layer attached to the substrate. Indeed, when flowed through the channel at high flow rates, the reactants have less time to diffuse and to generate aryl radicals that can subsequently react with the HOPG substrate. Moreover, at high flow rates, the generated aryl radicals go over the substrate at concentrations lower than when they are flowed at low flow rates. This is confirmed by the results of numerical simulations, which showed that the concentration of aryl radicals decreases from around 3 $\mu$M at 50 $\mu$L min$^{-1}$ to 2.5 $\mu$M at 200 $\mu$L min$^{-1}$ (Supplementary Fig. 23). The decrease in aryl radical concentration for flow rates of 50 to 200 $\mu$L min$^{-1}$ is in line with the decrease in layer thickness that is observed experimentally in this range of flow rates (Fig. 5b). On the other hand, the corral diameter slightly increased from 60 nm to 80 nm and the number of corrals generally decreased when the flow rate was increased. For instance, the density of corrals decreased from 30 corrals/$\mu$m$^2$ to around 10 corrals/$\mu$m$^2$ when the flow rate increased from 50 $\mu$L min$^{-1}$ to 200 $\mu$L min$^{-1}$ (Fig. 5c). These results are consistent with our previous observation that lower concentrations of aryl radical near the HOPG substrate imply larger the corral size and smaller corral density (comparing KI and NBD sides in Fig. 3c, d).

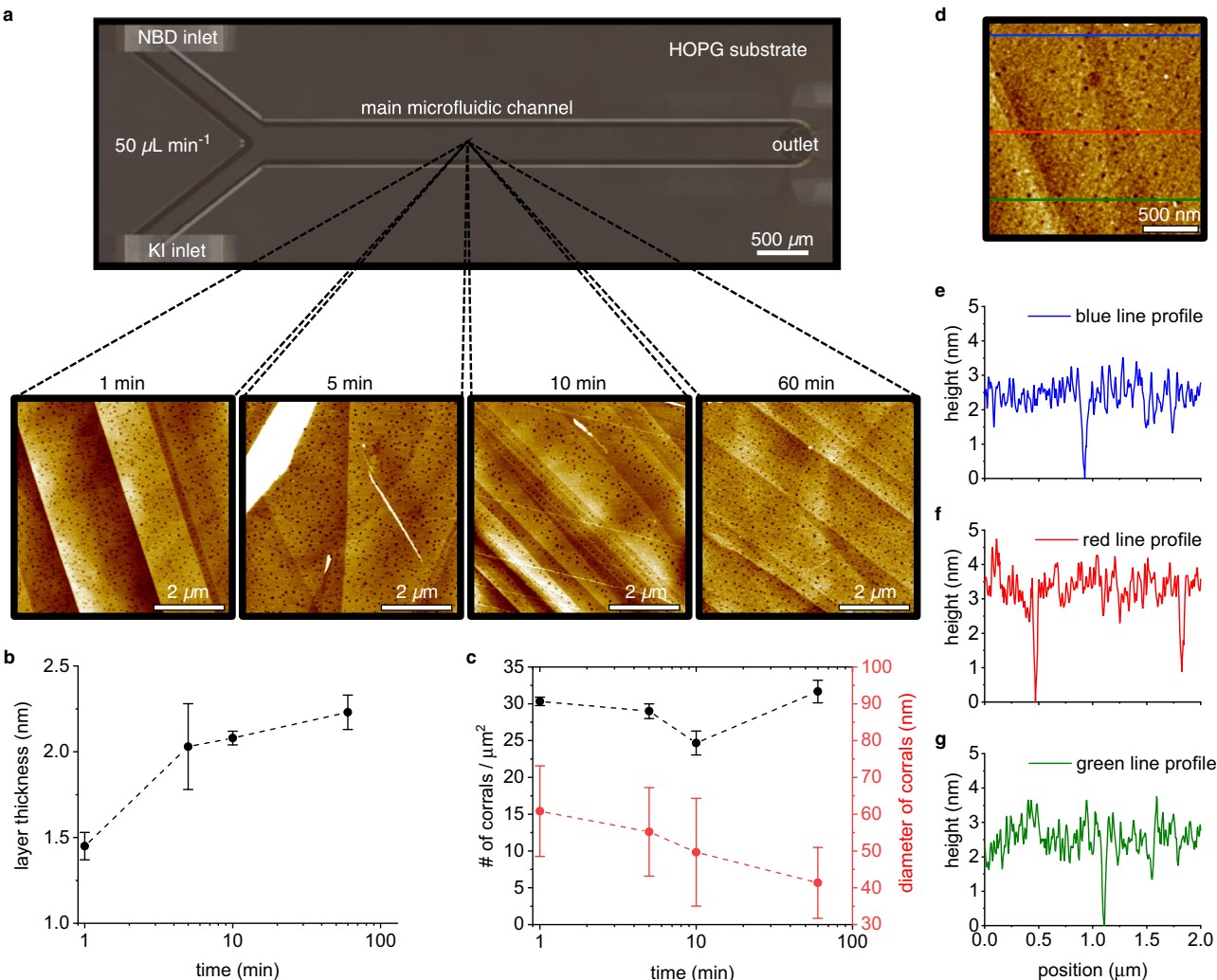

**Fig. 4 | Effect of flow-time on the grafted layer. a** Schematic top view of the microfluidic channel showing the inlets and outlet ports. The dashed lines indicate the location (i.e. the center of zone 2) where the AFM images were acquired. Panel **a** also includes the AFM images of the grafted layers obtained with flow times ranging from 1 to 60 min at a fixed flow rate of 50 μL min⁻¹. Scale bars in **a**: 2 μm. **b** Graph obtained from the analysis of the AFM images acquired at zone 2. The thickness of the grafted layer increases with the flow time. **c** shows how the number (black) and diameter (red) of corrals vary with respect to the flow time. Error bars represent the standard deviation of at least three different measurements. **d–g** AFM image (in **d**) indicating the sections used to generate the three topographic profiles presented in **e**, **f**, and **g**. The results presented in **d–g** was obtained from the experiment performed with a flow rate of 50 μL min⁻¹ for 120 min. Scale bar in **d**: 500 nm. Note that time dependent experiments were separately performed for each time point and the grafted layers on HOPG substrates prepared with different experimental times were independently characterized.

Even though we demonstrated that increasing the flow rate in our microfluidic approach reduces the thickness of the aryl grafted layer, we clearly observed that a flow rate of 200 μL min⁻¹ is not sufficient to induce the formation of a monolayer (i.e. a 1.0 nm-thick layer), with a dendritic growth still being apparent given the layer thickness of 1.2 nm. To demonstrate the control offered by our microfluidic approach, we further increased the flow rate to 600 μL min⁻¹ and observed the formation of a grafted layer of only 1.0 nm, even after a longer flow time of 10 min (Fig. 5d–g). This result further supports the observation that higher flow rates of the reactants imply thinner layers of grafted aryl radicals. Moreover, this is also consistent with having a fast surface attachment of the first layer and a slow dendritic growth of the multilayer, as observed when varying the flow time in the experiments presented in Fig. 4. The data obtained show that a flow rate of 600 μL min⁻¹ is sufficient to enable the grafting of a first layer of the aryl radicals generated in the main microfluidic channel, but it does not allow for the occurrence of the slower dendritic growth. Strikingly, a flow rate of e.g. 10 μL min⁻¹ also resulted in the formation of a mono-layer within 10 min (Supplementary Fig. 24). This is because the higher

production of aryl radicals at low flow rates is compensated for by the higher deactivation via the reaction with I₂ (see Supplementary Notes 2 and 5). Our numerical simulations—which consider this interplay between production and deactivation—confirmed that the concentration of aryl radical at 10 μL min⁻¹ is similar to that at 600 μL min⁻¹ (Supplementary Fig. 23); a result clearly explaining the formation of a monolayer for both flow rates. However, despite the similar aryl radical concentrations and layer thicknesses at 10 and 600 μL min⁻¹, the monolayers generated have quite different morphology (see Supplementary Fig. 24b and Fig. 5d, respectively). For example, the number of the corrals generated at 10 μL min⁻¹ is dramatically lower during the formation of the first layer, because of the much higher time for the RD process. This illustrates that the flow rate can be leveraged to independently control the layer thickness and morphology using our microfluidic approach. Next, we further increased the number of radicals passing over the HOPG substrate by simply flowing the two reactants at 10 μL min⁻¹ but for a longer time, i.e. during 60 min. This longer reaction time allowed for the growth of a densely-packed grafted layer of around 2 nm, a thickness that approximately equals

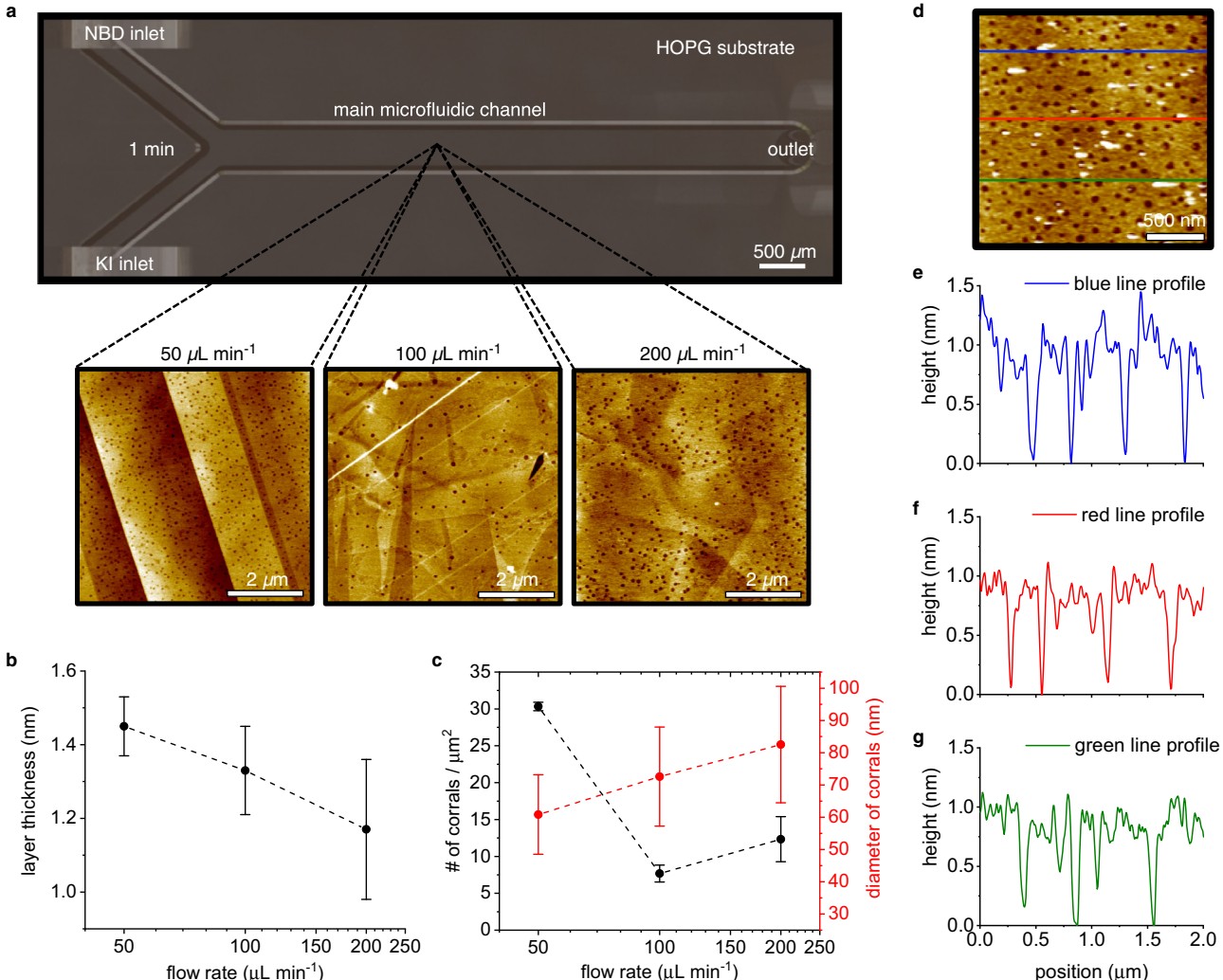

**Fig. 5 | Impact of flow rate on the grafted layer. a** Schematic top view of the microfluidic channel showing the inlets and outlet ports. The dashed lines indicate the location (i.e. zone 2) where the AFM images were acquired. Panel **a** also includes the AFM images of the grafted layers obtained at different flow rates (50, 100 and 200 μL min⁻¹) with a fixed flow time of 1 min. Scale bars in **a**: 2 μm. **b** The graph shows the decrease in layer thickness observed with increasing flow rate. **c** Graph showing the change in number (black) and diameter (red) of corrals with respect to the flow rate. Error bars represent the standard deviation of at least three different measurements. **d**–**g** AFM image (in **d**) indicating the sections used to generate the three topographic profiles presented in **e**–**g**. The results presented in **d**–**g** was obtained from the experiment performed with a flow rate of 600 μL min⁻¹ for 10 min. Scale bar in **d**: 500 nm.

the length of two aryl moieties covalently linked in a layer-by-layer fashion. Note that the morphology of this 2-nm thick layer is different from that of the other samples showcasing a dendritic growth, i.e. no corrals are found in this layer (Supplementary Fig. 25).

The results described above show that our microfluidic approach offers the possibility to generate chemical gradients onto HOPG with 2D and 3D control. As shown in Supplementary Table 2, we have demonstrated that at medium flow rates (50 μL min⁻¹), multilayer grafted layers of aryl moieties can be generated and their thickness can be tailored from 1.5 nm to 3.0 nm by adjusting the flow time from 1 minute to 120 min. Moreover, we have also observed that lower (10 μL min⁻¹) and higher (600 μL min⁻¹) flow rates induce the formation of a monolayer (1 nm). This controlled functionalization of a graphenic surface is highly relevant in the surface science field as it can lead to new functions and applications. For example, the layer thickness, size and density of corrals can strongly influence the on-surface properties of HOPG, such as its hydrophilicity. Additionally, the spatially-controlled grafting can also be explored to fine-tune the interaction with external adsorbates, and this can be relevant in the context of developing new sensing technologies[34]. The spatially-controlled

grafting of aryl moieties onto HOPG can also provide special experimental conditions and opportunities, e.g. to unveil the impact of nanoconfinement on the formation of self-assembled molecular networks in one go by employing one of our microfluidic grafted HOPG substrates[35]. Note that a variety of different nanoconfinement conditions can be generated in a single substrate via the controlled gradient formation prevailing inside our microfluidic device. Besides, we have also demonstrated the capacity of our method in establishing and transferring chemical gradients onto an underlying substrate via an alternative chemical approach, i.e. generating chemical gradients of a thiolated molecule onto a gold substrate; a result that clearly indicates the efficiency of our approach for spatially controlled functionalization of a targeted substrate (see Methods and Supplementary Fig. 26).

In summary, we have shown that the no-slip boundary condition present in continuous flow microfluidic devices allows for a covalent derivatization of HOPG with an exquisite spatiotemporal control, both in 2D and 3D. With the microfluidic approach, we were able to control the morphology (i.e. corral size and distribution) and thickness of the aryl grafted layer simply by taking advantage of the flow conditions (e.g. flow rate and flow time) inside the microfluidic device to change

the dynamics of the transport processes during the functionalization of HOPG. Note that these parameters cannot be adjusted via conventional bulk derivatization methods.

Our microfluidic approach has the potential to advance not only the controlled covalent modification of HOPG, but also the functionalization of other technologically-relevant substrates where spatially-controlled surface chemistry will be essential for the engineering of new functionalities and advanced devices.

## Methods

### Materials

4-Nitrobenzenediazonium (NBD) tetrafluoroborate (97%), potassium iodide (KI) (99.9%), Deuterium oxide (99.9 atom%), and 1-dodecanethiol (≥98%) were purchased from Sigma-Aldrich. Ethanol (spectroscopy purity) was bought from Uvasol. All chemicals were used without further purification. High purity water (Milli-Q, Millipore, 18.2 MΩ cm) was used for all the experiments. The highly ordered pyrolytic graphite (HOPG, grade ZYB, Advanced Ceramics Inc., Cleveland, USA) substrates were freshly cleaved before use. The home-made flow cells were machined with Poly(methyl methacrylate) (PMMA) and Polyether ether ketone (PEEK). The glass syringes (HAMILTON, 5.0 mL) and the microfluidic pump (Harvard Apparatus, Pump 11 Elite) were used to perform the microfluidic system with precise control of flow rate and flow time. Microchannel-to-world interphase was achieved by using microfluidic connectors (10-32 Coned for 1/16″ OD, IDEX Health & Science, LLC, USA) connected to PTFE tubing (1/16″ OD, IDEX Health & Science, LLC, USA).

### Design and fabrication of the microfluidic cell

The microfluidic device was designed using a 3D computer-aided design (CAD) software (SOLIDWORKS 2018) to provide a laminar flow of the two reactant solutions over the HOPG substrate. The microfluidic device consists of two layers machined either from poly(methyl methacrylate) (PMMA) or polyether ether ketone (PEEK). While a PMMA layer enables to observe what happens on the HOPG surface due to its transparency, the PEEK one improves the chemical resistance of the microfluidic device (Supplementary Fig. 2a, b). The top layer includes a Y-shaped microchannel with a 400 μm-wide and ~7 mm-long main channel, together with input/output ports for the microfluidic connectors (10-32 Coned for 1/16″ OD, IDEX Health & Science, LLC, USA) to connect PTFE tubing (1/16″ OD, IDEX Health & Science, LLC, USA). The bottom layer acts as a support to press a square shaped (12 mm × 12 mm) HOPG substrate onto the top layer. The tinny groove (50 μm in width and 50 μm in height) machined around Y-shaped microfluidic channel on the top layer not only behaves as a sealing element to enclose the reaction area and avoid leakage, but also defines the height of the confined gap (120 μm) which is specifically the sum of the heights of the sealing groove (50 μm) and the Y-shaped microchannel (70 μm).

### Microfluidic experimental procedure

The microfluidic experimental procedure was divided into seven steps. (i) The microfluidic device was washed with milli-Q water and dried with air gas flow. Once cleaned, it was checked using optical microscopy; (ii) A fresh graphitic surface (clean and flat) was prepared by peeling off the top layers of HOPG using an adhesive tape; (iii) The HOPG sample was placed into the cleaned microfluidic device, and then the corresponding bottom layer was put onto the back surface (not cleaved) of HOPG. Six screws were used to make sure that the top and bottom layers were tightly clamped with the HOPG between them. Note that the fresh surface of HOPG faced the microfluidic network present in the top layer; (iv) The two inlets and one outlet of the top layer were connected to PTFE tubes. The microfluidic device was continually injected with a stable air gas flow from one of the connected tubes, and then the microfluidic device was immersed into water to check for eventual gas leakage. In case of gas leakage, the microfluidic device would be further tightened by readjusting the screws once more. Note that the microfluidic network was always kept in dry conditions; (v) Milli-Q water was used to prepare the NBD and KI solutions, both with concentrations of 10 mM. Two syringes were used to inject the NBD and KI solution into the microfluidic device via the two independent inlet ports, ensuring that the gas or bubbles had been completely removed from the syringe; (vi) The pump was opened and set with the parameters used in each specific experiment. The parameters included the diameter of the syringe, flow rate, and flow time; (vii) When the pump automatically stopped, the microfluidic device was removed from the whole set-up. Next, it was opened to take out the HOPG substrate. The functionalized HOPG was then washed with ethanol and Milli-Q water. Finally, the HOPG was dried with argon gas and used for future characterizations.

### Diffusion ordered spectroscopy – Nuclear magnetic resonance (DOSY-NMR)

The DOSY-NMR measurements were performed with a Bruker Avance II + 600 NMR spectrometer. A total of 3 mg NBD was dissolved in about 0.6 mL $D_2O$ for 20 mM NBD solution. The $^1H$ DOSY (Diffusion Ordered Spectroscopy) spectra were recorded based on the 20 mM NBD solution.

### Ultraviolet visible (UV–vis) spectroscopy

UV-visible absorption spectroscopy was performed with Cary 60 UV–visible spectrophotometer (Agilent Technologies). NBD and KI aqueous solutions were mixed together at appropriate mole ratios and the mixed solution was quickly transferred to a 1 mm optical path quartz cuvette. The reaction was monitored by acquiring 20 s long spectra from 400 to 200 nm every 2 min.

### Atomic force microscopy (AFM)

AFM imaging was performed with a Cypher ES (Asylum Research) system at 32 °C in tapping mode at the air/solid interface. OMCL-AC160TS-R3 probes (spring constant ~26 N/m) with a resonance frequency around 100 kHz were used. OMCL-AC240TS-R3 probes (spring constant ~2 N/m) with a resonance frequency around 70 kHz were used for scratching the functionalized layer.

### Sample positioning for AFM analysis

Sample positioning for AFM imaging was performed with optical microscopy. The clamping of the HOPG with the microfluidic device left a mark from the channel grooves on the HOPG sample, which is visible under an optical microscope (Supplementary Fig. 2e). Based on the width and length of the groove marks, the AFM tip was accordingly positioned for the topography characterization. For a specific position, an optical and AFM image were sequentially collected. The optical image was used to identify the position of the corresponding AFM image in the microfluidic network machined in the top layer. The location was determined by two coordinates, respectively: latitudinal coordinate ($x$) determined by width of groove mark and longitudinal coordinate ($y$) determined by length of groove mark.

### AFM measurements (analysis for layer height, corral size and distribution)

The AFM images were analyzed using SPIP 6.3.5 to obtain statistics about relevant parameters, i.e. layer thickness, corral number, size, and diameter. The corrals in the functionalized layer were marked, to determine their number, size and diameter (see the example in Supplementary Fig. 18). Line profiles were used to identify the layer thickness based on the corral (empty) area (see the example in Supplementary Fig. 19). In order to identify the bottom of the empty corral as the HOPG surface, AFM tip scratching experiments were performed by using the contact scanning mode of AFM with a force of about

80 nN to locally remove the functionalized layer and expose the bare HOPG surface. The AFM tip scratching with a force of about 80 nN has been shown to have no morphological impact on the bare HOPG surface[36]. Three AFM images (1 μm by 1 μm) were used for each location in the channel area. The average result and the corresponding standard deviation were calculated based on the three groups data for each location in the channel area.

## Raman spectroscopy

Raman experiments were performed at room temperature (21–23 °C) by using a Raman microscope (Monovista CRS + , S&I GmbH). The 632.8 nm He–Ne laser light was directed and focused onto the sample surface by using an objective (OLYMPUS, BX43 100×, N.A. 0.7). The optical density at the sample surface was about 590 kW cm$^{-2}$. Raman scattering was collected with the same objective and directed to a Raman spectrograph (S&I GmbH) equipped with a cooled charge-coupled device (CCD) camera operating at −100 °C (Andor Technology, DU920P-BX2DD). Accumulation time for all spectra was 6 s.

## Raman measurements (calculation of I(D)/I(G))

Similar to the AFM imaging, the positioning of the Raman measurements was based on optical imaging. In each specific Raman spectrum, the D band (-1330 cm$^{-1}$) and G band (-1580 cm$^{-1}$) were focused. The intensity of the D band (I(D)) and the intensity of G band (I(G)) were used to calculate the I(D)/I(G) ratio (see Supplementary Fig. 21).

## Transferring the chemical gradient of a thiolated molecule onto the gold substrate

Reactive gold (Au) substrates were prepared by coating the silicone (Si) wafer with chromium (Cr) and Au thin films to create the adhesion and reactive layers, respectively. Both Cr adhesion layer and Au coatings were deposited with DC sputtering (vonAdrenne CS 320 S). First, Si substrate was cleaned with acetone and isopropyl alcohol (IPA) via ultrasonication for 3 min each, respectively. Subsequently, 10 nm Cr adhesion layer was deposited with DC power 0.25 kW under 0.006 mbar Ar atmosphere. Finally, 100 nm Au layer was deposited with DC power 0.2 kW under 0.006 mbar Ar atmosphere. After the metallization process, Au-coated Si wafer were diced into square pieces (25 mm × 25 mm) to be utilized as a reactive substrate in the microfluidic experiments. A two inlet (Y-junction) polydimethylsiloxane (PDMS)-based microfluidic device was mechanically clamped against the Au-coated Si substrate for sealing the channels. To accomplish the gradient of thiolated molecule within the main channel (i.e. 300 μm in width) and transfer it onto gold substrate, the pure ethanol and an ethanolic solution of 1-dodecanethiol ≥98% (15 mM) were injected (at a flow rate of 10 μL min$^{-1}$ for 30 min) into the main channel using two different inlets of the Y-junction microfluidic device. By co-flowing the pure ethanol and ethanolic solution of 1-dodecanethiol, a concentration gradient of thiol molecule was obtained along the main channel's width (Supplementary Fig. 26a). After 30-min experimental time, the microfluidic device was flushed with the pure ethanol (at a flow rate of 100 μL min$^{-1}$ for 2 min) to remove excess thiol solution from the channel and over the gold substrate. Finally, the gold substrate was dried under N$_2$ gas and characterized using AFM. To quantify the transferred gradient onto the gold substrate, the root mean square height analysis were performed on AFM images obtained from different locations (i.e. along the channel width) of the covalently modified substrate Supplementary Fig. 26b, c).

## Numerical simulations

To investigate the concentration gradients forming over the HOPG substrate in the microfluidic device, we simulated the flow and mass transport in the microfluidic device using a computational fluid dynamics approach based on the finite volume method (in line with previous works simulating the flow and mass transport inside microfluidic devices)[37–39]. The steady-state velocity, pressure, and species concentration were calculated by coupling the Navier–Stokes equation for an incompressible Newtonian fluid, the continuity equation, and the species transport equation, which were respectively given by:

$$\frac{\partial \vec{\mathbf{V}}}{\partial t} + \vec{\mathbf{V}}(\nabla \cdot \vec{\mathbf{V}}) = -\frac{1}{\rho}\nabla P + \upsilon \nabla^2 \vec{\mathbf{V}} \tag{1}$$

$$\frac{\partial \rho}{\partial t} + \nabla(\rho \vec{\mathbf{V}}) = 0 \tag{2}$$

$$\frac{\partial(\rho Y_i)}{\partial t} + \nabla\left(\rho \vec{\mathbf{V}} Y_i\right) = \rho D_i \nabla^2 Y_i + R_i \tag{3}$$

where $\vec{\mathbf{V}}$ is the velocity vector, $\nabla$ is the divergence operator, $\rho$ is the fluid density, $P$ is the pressure, $\upsilon$ is the kinematic viscosity, $\nabla^2$ is the Laplacian operator, $Y_i$ is the mass fraction of species $i$, $D_i$ is the diffusion coefficient of species $i$ and $R_i$ is the reaction source/sink of species $i$. The volumetric reaction rate for aryl radical production was considered to be described by:

$$R_{\text{radical}} = 0.13 \cdot [\text{NBD}]^{0.62} \cdot [\text{KI}]^{0.435} \tag{4}$$

where [NBD] and [KI] are the molar concentration of NBD and KI, respectively. The rate of the reaction with the graphitic surface, i.e., the rate of formation of the grafted layer, was assumed to be described by:

$$R_{\text{grafting}} = k_1 \cdot [\text{NBD}] \tag{5}$$

where $k_1$ is the substrate reaction rate constant of the grafting step, and [NBD˙] is the aryl radical molar concentration. The rate of production of iodine (I$_2$) via the reaction between two iodine radicals (I˙), was assumed to be described by:

$$R_{\text{iodine}} = k_2 \cdot [\text{I}]^2 \tag{6}$$

where $k_2$ is the rate constant for the formation of iodine, and [I˙] is the concentration of iodine radical. Based on literature[27], the rate of deactivation of the aryl radical by reaction with iodine was described:

$$R_{\text{deactivation}} = 10^{10} \cdot [\text{NBD}] \cdot \left[\text{I}_2\right] \tag{7}$$

In the four reactions above, the reaction rate constants $k_1$ (grafting step) and $k_2$ (iodine formation step) were unknown. For that reason, we tested different values of $k_1$ and $k_2$ to identify $(k_1, k_2)$ pairs that could generate curves of aryl radical concentration along the main reactor (both along its length [y axis, Fig. 2d] and its width [x axis; Fig. 2f]) that are consistent with the experimentally-obtained thickness of the grafted layer (Fig. 3b). We found that $k_1 = 10^{-3}$ m s$^{-1}$ and $k_2 = 2 \times 10^5$ M$^{-1}$ s$^{-1}$ fitted the experimental data for all the tested flow rates, and therefore these rate constants were used in simulations (see details in Supplementary Note 5).

Fluid properties were assumed to be those of water ($\rho = 1000$ kg m$^{-3}$; μ = 0.001 Pa·s) given that the solutions are diluted. The diffusion coefficients of NBD was assumed to be $8.14 \times 10^{-10}$ m$^2$ s$^{-1}$ (determined by DOSY-NMR, Supplementary Note 1) and that of KI was assumed to be $2 \times 10^{-9}$ m$^2$ s$^{-1}$ based on literature[26]. The 3D domain of the microfluidic flow-cell was finely meshed to minimize the occurrence of non-physical artificial diffusion (Supplementary Fig. 5a), and to capture accurately the transport processes and the surface reaction near the HOPG. A mesh containing -13 million cells (built with -400 cells along x, 550 cells along y and 60 cells along z) was used in simulations, as it was found to ensure accurate

simulation of the flow and mass transport inside the device. This was confirmed by comparing the predictions of the present model against analytical solutions from independent literature sources. Specifically, we compared the predicted velocity profile along the channel width with the corresponding analytical solution (Supplementary Fig. 3a), the predicted NBD concentration profile along the channel width with the corresponding analytical and numerical 1D solutions (Supplementary Fig. 3b), and the predicted aryl radical concentration along the reactor with the corresponding analytical solution (Supplementary Fig. 4). The fact that the predicted velocity and concentration profiles are consistent with the corresponding analytical solutions indicates that the present model is well implemented, and, thus, can be used to accurately predict the flow and mass transport in microfluidic devices. The boundary conditions at the inlets were chosen to match the experimental conditions (i.e. the flow rate and concentration) and the outlet boundary was assumed to be at atmospheric pressure. No-slip boundary condition was considered at the walls of the microchannel, and the surface reaction was assumed to occur only on the HOPG. A steady-state, double precision, pressure-based solver was used, considering second-order discretization, and the SIMPLE algorithm was used for velocity-pressure coupling. Convergence was assumed when residuals were less than $10^{-5}$, with stricter criteria producing similar concentration and velocity fields.

## Reporting summary

Further information on research design is available in the Nature Portfolio Reporting Summary linked to this article.

## Data availability

The raw data that support the results of this study are available from the corresponding authors on request.

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

## Acknowledgements

J.P.L. acknowledges the European Research Council Starting Grant microCrysFact (ERC-2015-STG No. 677020), the Horizon 2020 FETOPEN project SPRINT (No. 801464), the Swiss National Science Foundation (project no. 200021_181988), grant PID2020-116612RB-C33 funded by MCIN/AEI/10.13039/501100011033. J.P.L. and T.S.M acknowledge support from the EU, from the Horizon 2020 FETOPEN project SPRINT (No. 801464). S.D.F. acknowledges the Fund of Scientific Research Flanders (FWO), and KU Leuven-Internal Funds. This work was also in part supported by FWO under EOS 30489208. T.S.M. and J.P.V. acknowledge the support by LA/P/0045/2020 (ALiCE), UIDB/00532/2020 and UIDP/00532/2020 (CEFT), funded by Portugal through FCT/MCTES (PIDDAC). Y.X. acknowledges financial support through the China Scholarship Council (CSC) (201706890021). J.S. acknowledges financial support through a Marie Skłodowska-Curie Individual Fellowship (EU project 789865-EnSurf). We thank Dr. David E. Clarke for his support with DOSY measurements to determine the diffusion coefficients of NBD and Dr. Kunal S. Mali for fruitful discussions.

## Author contributions

Y.X., S.S., and J.P.V. contributed equally to this work. S.S. designed the microfluidic experiments and microfluidic device. J.P.V. performed the numerical simulations. Y.X. and J.S. performed the HOPG grafting experiments and related characterizations. S.S. and Y.X. analyzed and interpreted the experimental data. D.R.S.M determine the kinetics of radical formation using UV–vis data. D.K. and S.P. prepared the gold coated silicone substrates and performed experiments on the gold substrate. J.P.V. and T.S.M. interpreted the numerical simulation results. S.D.F. and J.P.L. conceived the idea. All authors contributed to the writing of the manuscript.

## Competing interests

The authors declare no competing interests.
