## [Peer Review File · Nature Communications]

Reviewer comments, first round review –

Reviewer #1 (Remarks to the Author):

This is a very interesting paper that brings new opening in the surface functionalization of HOPG by diazonium salts. The study is very detailed and the authors succeed in preparing a monolayer of aryl groups which is a much desired feature in the field. The conclusions are correlated by a number of experiments: simulations, AFM, Raman. Therefore the paper can be published taking in account the minor remarks quoted below.

Some Remarks.

1 A similar investigation has already published (although with less details) it must be absolutely quoted: Surface Patterning Using Two-Phase Laminar Flow and In Situ Formation of Aryldiazonium Salts. Andrew J. Gross, Volker Nock, Matthew I. J. Polson, Maan M. Alkaisi, and Alison J. Downard* Angew. Chem. Int. Ed. 2013, 52, 10261 –10264, 2013.

2 The authors refer to the formation of corrals, but there are really holes in the layer and this is a main drawback of this reaction. Of course PVD gold on Si is not monocrystalline or at least the terraces are much smaller, however it is less hydrophobic than HOPG and such holes should not exist. It is possible to obtain larger terraces on mica, but this material would be difficult to tight in the set up. Nevertheless a comparison with Si-Au would be interesting.

3 The authors consider give the rate of the grafting reaction as k_1 of $10^{-3} \text{ m}\cdot\text{s}^{-1}$; This is an important value that should be included in the text as, to the best of my knowledge this value is unknown

4. 10 mM is a quite high concentration for grafting 4-NBD, there is possibly formation of oligomers in solution. Are they observed?

5. Fig S6 radical production (purple blue)

6. Considering the corrals on the KI side. These corrals are large due to the low concentration of aryl radicals, this means that there is a some stacking effect; the incoming radical prefers to graft next to an already grafted aryl group than in the middle of an empty space?

"This result can be explained by the lack of control over the mass transport of aryl radicals onto the HOPG substrate in time and space during the grafting process." But also by the local reactivity of the radical

7. line 234. "Clearly, with 234 these experiments, we proved that even though the reaction of aryl radicals with a previously generated 235 aryl grafted layer is slower than the direct reaction of the aryl radicals with the HOPG, .." I am not convinced by this sentence , which, I think, is in contradiction with STM images of 4-nitroaryl groups grafted on HOPG where one observed grafted dimers and empty space which means than the incoming radical prefers to attach the first grafted group rather than an an empty spot on HOPG (ACS Nano 2015)

Reviewer #2 (Remarks to the Author):

This manuscript (Covalent transfer of chemical gradients onto a graphenic surface with 2D and 3D control) presents a microfluidics method to generate controlled chemical gradients onto a graphenic material with 2D and 3D control. As the novelty of this study has not been clearly stated in this study, many fundamental questions should be well addressed. Below is the list of questions and suggestions.

Major comments:

1. In the Introduction, the background and significance of the study are unclear. I suggest that the authors revise the Introduction.

2. The Y-shaped microchannel is commonly used in microfluidic studies, which is the simplest

configuration enabling the generation of compositional gradients between two reactants injected through the two inlet ports. Moreover, the laminar flow, no-slip boundary conditions and mixing by molecular diffusion have been deeply studied by researchers of fluid mechanics.

3. The microfluidic method for controlling the chemical gradients has been reported in many reports.

4. The background physics and mechanism for the generation of HOPG have not been deeply studied.

5. The rationality of the numerical simulation method and the reliability of the simulation results are not proved.

6. The innovation of this study should be clearly explained.

Reviewer #3 (Remarks to the Author):

The authors proposed one method to well control the functionalization of graphenic materials by continuous-flow microfluidic devices. This is a detailed report on the device design and the control experiment for the grafting density, layer thickness and morphology of the functionalized graphenic materials, which can be tuned by flow time and flow rate. By combining the numerical simulation with experiments, the manuscript presents the ability to control the grafting process of graphenic materials and tried to figure out the physical mechanism behind the phenomena. This is an interesting paper that will attract some attention from the researchers working on the functionalization technology, while the following specific issues should be improved:

1. The authors highlighted that the better controlling during the microfluidic grafting in the proposed microfluidic devices is thanks to the no-slip boundary condition, both in the introduction and also in the Results. However, they just showed an indirect evidence that the concentration of intermediate product aryl radical is larger at the top and bottom walls by numerical simulation to support this proposal. Can the author present more direct experimental evidence for this claim? If the no-slip boundary condition doesn't work, what would the morphology of grafted HOPG be like? Is it possible to present the experimental results of this control experiment? In addition, what is the limited length for the no-slip boundary condition to work?

2. From the numerical simulation, the RD area can cover at least 300 μm in zone 2. In experiments, what is the size of the effective RD area (well-controlled morphology of grafted HOPG) at zone 2? From Fig. 3a, the morphology is not so homogenous along the x-axis. The size data of effective RD area would be what the readers care about and useful to evaluate the significance of the proposed devices.

3. Figure 4a presents the experimental results at a fixed position with varying time. I am confused why the morphology (stripes) of HOPG changes in the four pictures a lot.

4. How does the author define the basal height of the HOPG to obtain the layer thickness of the grafted layer?

5. The authors have changed parameters, such as the flow time, flow rates, to see how the grafting process occur on the HOPG by identifying the thickness of grafted layer, size and density of corrals. Upon changing these parameters, the morphology of the grafted HOPG changes a lot. I suggest the author to summarize the optimal condition to obtain the monolayer and bilayer homogenous grafted layer on the HOPG at the end of the Results and discussions. Additionally, how do the thickness of grafted layer, size and density of corrals of HOPG affect its functionality?

6. In this manuscript, the author compared the morphology between the proposed microfluidic devices and those obtained from bulk method. Does the functionalized graphenic materials obtained in this manuscript work as well as those obtained from the bulk method, or much better? Can the authors add some control experiments?

Our detailed answers to the points raised during the review process are included below.

Reviewers' comments:

Reviewer #1:

This is a very interesting paper that brings new opening in the surface functionalization of HOPG by diazonium salts. The study is very detailed and the authors succeed in preparing a monolayer of aryl groups which is a much desired feature in the field. The conclusions are correlated by a number of experiments: simulations, AFM, Raman. Therefore the paper can be published taking in account the minor remarks quoted below.

We thank Reviewer #1 for these positive comments and for highlighting and appreciating the novelty of our approach as well as of the results obtained.

Some Remarks.

1. *A similar investigation has already published (although with less details) it must be absolutely quoted: Surface Patterning Using Two-Phase Laminar Flow and In Situ Formation of Aryldiazonium Salts. Andrew J. Gross, Volker Nock, Matthew I. J. Polson, Maan M. Alkaisi, and Alison J. Downard* Angew. Chem. Int. Ed. 2013, 52, 10261 –10264, 2013.*

We agree with the reviewer, and we have now included this reference in the main text. However, we would like to emphasise that while Downard *et al.* employed a polydimethylsiloxane (PDMS)-based continuous flow microfluidic device to pattern [p-(dimethyltriaz-1-en-1-yl)phenyl]methylamine (AMP) on a pyrolyzed photoresist film (PPF), it should be noted that the authors could not achieve a good control over the grafting density, the layer thickness and/or morphology. In fact, Downard *et al.* demonstrated that laminar flows can be used for patterning AMP films, however, the size of the patterned films was limited to a small area of the main microfluidic channel. That is, in the work indicated by Reviewer #1, the controlled patterning of AMP only occurs in a narrow region of the main microfluidic channel.

2. *The authors refer to the formation of corrals, but there are really holes in the layer and this is a main drawback of this reaction. Of course PVD gold on Si is not monocrystalline or at least the terraces are much smaller, however it is less hydrophobic than HOPG and such holes should not exist. It is possible to obtain larger terraces on mica, but this material would be difficult to tight in the set up. Nevertheless a comparison with Si-Au would be interesting.*

We thank the reviewer for his/her suggestion. We have now performed additional experiments employing a Y-junction PDMS-based microfluidic device to control the functionalization of gold substrates with 1-dodecanethiol. In sharp contrast to the surface functionalization of HOPG with diazonium salts (where corral formation is always occurring due to the generation of nitrogen), it is well-known that the functionalization of gold substrates with thiolated molecules can led to perfectly uniform self-assembled monolayers (SAMs), see e.g. *Chemical Reviews, 2005, 105(4), 1103-1169*. Therefore, as highlighted by Reviewer #1, because that type of reaction is not based on a radical grafting process, it will enable the generation of chemical gradients onto gold substrates, without formation of "holes" or corrals. Indeed, the functionalization of a gold substrate with our microfluidic approach can be ideal to directly

demonstrate the capacity of covalently transferring chemical gradients with a different (alternative) chemical approach. Note that in bulk functionalization experiments of gold substrates with thiolated molecules, it is extremely challenging to covalently transfer chemical gradients of the thiolated molecules onto the gold substrate. We thus thank very much the reviewer and the editor for suggesting these set of experiments as they clearly showed the great efficiency of our microfluidic device in establishing and transferring chemical gradients onto an underlying substrate, via the controlled mixing achieved inside the microfluidic channel, where laminar flow prevails and only diffusion plays a key role during the functionalization.

Fig. R1. Gradient formation on Si-Au. **a**, Schematic drawing of PDMS-based microfluidic device and experimental configuration. **b**, AFM images showing the gradient of 1-dodecanethiol over Si-Au substrate. From left to right: position (1), outside of the main microfluidic channel (left side); positions from (2) to (8), locations measured along the width of the main microfluidic channel (300 μm) and from the pure EtOH-laden flow towards the 1-dodecanethiol-laden flow. Note that the concentration of 1-dodecanethiol increases when moving towards the 1-dodecanethiol-laden flow; and position (9) outside of the main microfluidic channel (right side). Scale bars: 2 μm , 0.5 μm , 0.2 μm for top, middle and bottom rows, respectively. **c**, Root mean square (Sq) height calculated from AFM images with respect to relative position for quantification of gradient formation under the main channel area. The error bars correspond to the standard deviation of the experimental results, for a $n=3$.

As shown in **Fig. R1**, we generated a chemical gradient of 1-dodecanethiol inside our microfluidic device by co-flowing pure EtOH and 1-dodecanethiol (15 mM in EtOH) at a flow rate of $10 \mu\text{L min}^{-1}$ (for each flow stream) and for 30 minutes (see **Fig. R1a**). One can clearly see that the gold substrate is less covered with 1-dodecanethiol molecules on the side of the

microfluidic channel where pure EtOH was injected (see **Fig. R1b**). However, when moving towards the side of the microfluidic channel where 1-dodecanethiol was injected, a denser covered gold substrate was manifest (see **Fig. R1b**). Note that, outside the microfluidic channel, mainly the bare gold substrate was observed, which clearly demonstrates the efficiency of our approach for covalently transferring the chemical gradients generated inside the microfluidic channel via a controlled diffusion. Undoubtedly, these additional data show that the no-slip boundary condition prevailing in our microfluidic device is key to control chemical gradients onto surfaces, as stated in our manuscript (see **Fig. R1c**). Additionally, it should be highlighted here that our approach is therefore not only working for the controlled functionalization of HOPG and that reactions other than a radical grafting can be employed.

3. *The authors consider give the rate of the grafting reaction as k_1 of $10^{-3} \text{ m}\cdot\text{s}^{-1}$; This is an important value that should be included in the text as, to the best of my knowledge this value is unknown.*

Following the reviewer's suggestion, we now indicate in the main text of the manuscript (see caption of **Fig. 2**) the rate constant of the grafting reaction (k_1), as well as that of the I_2 formation reaction (k_2), which influences the deactivation of the aryl radical. Furthermore, we have now included in the manuscript a new section (**Supplementary Information 3.6**) describing the parametric analysis conducted with 28 different combinations of k_1 and k_2 (**Supplementary Table 1**), to assess their effect over the concentration of aryl radical along the microfluidic device and identify the values of k_1 and k_2 fitting better the experimentally observed layer thicknesses.

4. *10 mM is a quite high concentration for grafting 4-NBD, there is possibly formation of oligomers in solution. Are they observed?*

Indeed, oligomers may form (see e.g., *J. Am. Chem. Soc.*, 1962, 84, 3847 or *Chem. -Eur. J.*, 2009, 15, 2101). Precipitates are observed in the 10 mM 4-NBD aqueous solution when the solution is stored at ambient conditions for several hours. Therefore, the 4-NBD solution was freshly prepared for each grafting experiment and the reaction time (flow time) was controlled within 2 hours.

5. *Fig S6 radical production (purple blue)*

We thank the reviewer for his/her suggestion. We have now corrected the text in the figure caption as "purple blue". Note that this figure is now **Fig. S14** in the revised manuscript.

6. *Considering the corrals on the KI side. These corrals are large due to the low concentration of aryl radicals, this means that there is a some stacking effect; the incoming radical prefers to graft next to an already grafted aryl group than in the middle of an empty space?*

"This result can be explained by the lack of control over the mass transport of aryl radicals onto the HOPG substrate in time and space during the grafting process." But also by the local reactivity of the radical.

Indeed, the sentence that Reviewer #1 refers to was included in the main text to explain the differences between the bulk experiments (i.e., drop-casting method, see **Fig. S20a**) and the experiments performed using our microfluidic approach (see **Fig. 3-5**). We would like to emphasize that the main difference between these two methodologies is that the latter benefits from the enhanced mass transport achieved inside the microfluidic channel to generate a spatially controlled grafting of aryl radicals onto HOPG. In sharp contrast, in the case of the bulk method, there is an absence of stable and controlled concentration profiles near the liquid-solid interface that led to non-uniform grafted areas with larger corrals. For further details, please see also our response to the 1st comment of Reviewer #3.

Additionally, we would like to point out that the reactivity differences experienced by neighbouring positions after the attachment of an aryl radical to a graphitic structure have been previously studied (see e.g. *How Do Aryl Groups Attach to a Graphene Sheet?*, *J. Phys. Chem. B*, 2006, 110, 23628-32 for the full discussion). In this study, it was found that there is an activation of the *para* positions of the aryl moieties that have been attached to the graphitic structure. In addition to this effect, the aryl diazonium-grafting process studied in our manuscript also encompasses the formation of nitrogen gas (see e.g., *Nanoscale*, 2020, 12, 11916–11926 and *ACS Nano*, 2019, 13, 5559–5571). Accordingly, these two effects contribute to the formation of the corrals observed.

7. *line 234. "Clearly, with these experiments, we proved that even though the reaction of aryl radicals with a previously generated aryl grafted layer is slower than the direct reaction of the aryl radicals with the HOPG, .." I am not convinced by this sentence , which, I think, is in contradiction with STM images of 4-nitroaryl groups grafted on HOPG where one observed grafted dimers and empty space which means than the incoming radical prefers to attach the first grafted group rather than an an empty spot on HOPG (ACS Nano 2015).*

The interfacial condition between the reaction solution and HOPG surface plays a very important role in the grafted layer growth by the radicals.

In the electrochemical grafting method (*ACS Nano* 2015, 9, 5, 5520–5535), the electric field mediates the diazonium cations to move to the surface of the cathode (HOPG), producing the aryl radicals near the surface of HOPG, so that the aryl radicals react with the surface where they are formed. As a result, previously grafted molecules can locally affect the formation of aryl radicals and thus the grafting layer, e.g., by an increased local concentration due to attractive interactions between already grafted aryls and the diazonium in solution or enhanced electron transfer and radical formation.

In reducing reagent (KI)-mediated grafting method, the produced aryl radicals are not concentrated near the HOPG surface as in the electrochemical method, but instead homogeneously distributed in the whole reaction solution. In the bulk drop-casting experiment, a static interface is formed between the reaction solution and the HOPG surface. The aryl radicals produced in solution can diffuse to the bare HOPG surface to form the first grafting layer, and simultaneously, additional diffusing radicals can graft to the previously formed grafting layer.

Remarkably, in the microfluidic channel, the distribution of the in-situ produced aryl radicals near the HOPG substrate is governed by the spatially controlled RD area that can be extended near the liquid-solid interface due to the no-slip boundary condition. Our experimental results demonstrate that with the microfluidic approach the reaction of aryl radicals with a previously generated grafted layer is slower compared to the direct reaction of the aryl radicals with the HOPG, both from the flow time-dependent experiment and flow rate-dependent experiment. The most apparent and direct proof is that the dendritic layer growth can be completely prevented at very high flow rates ($600 \mu\text{L min}^{-1}$), leading to the grafting of a monolayer. In other words, the dendritic layer growth is slowed down and even prevented at a high flow rate in our microfluidic experiment due to reduced RD time.

Accordingly, one should note that the grafting conditions achieved with our microfluidic approach are completely different from the conditions established with previously reported HOPG grafting experiments.

Reviewer #2:

This manuscript (Covalent transfer of chemical gradients onto a graphenic surface with 2D and 3D control) presents a microfluidics method to generate controlled chemical gradients onto a graphenic material with 2D and 3D control. As the novelty of this study has not been clearly stated in this study, many fundamental questions should be well addressed. Below is the list of questions and suggestions.

Major comments:

1. In the Introduction, the background and significance of the study are unclear. I suggest that the authors revise the Introduction.

We have modified the introduction in the revised version of our manuscript to provide more detailed information of the differences between our approach and other existing methods. Additionally, we also refer here to our answer to Reviewer #1's 1st comment.

2. The Y-shaped microchannel is commonly used in microfluidic studies, which is the simplest configuration enabling the generation of compositional gradients between two reactants injected through the two inlet ports. Moreover, the laminar flow, no-slip boundary conditions and mixing by molecular diffusion have been deeply studied by researchers of fluid mechanics.

We disagree with the above argument made by Reviewer #2. In fact, we think that using a simple Y-Junction geometry is one of the most convenient approaches to investigate the formation of controlled RD areas as well as the effect of the no-slip boundary condition during surface functionalization. The capability of the microfluidic device employed in the work presented is sufficient to clearly confirm the spatially-controlled grafting of aryl radicals to HOPG. We are pleased with the results obtained where the covalent transfer of chemical gradients onto a graphenic surface is demonstrated with in 2D and 3D control. Note that the results presented are unprecedented in the literature.

Additionally, we would like to emphasize that, while the geometry (Y-junction) seems simple to the Reviewer #2, the microfluidic devices presented in our work were specially designed to achieve the sealing of the entire microfluidic network via mechanical clamping over the HOPG substrates with a novel design approach (i.e., machined sealing grooves). The top layers were

machined either from PMMA or PEEK. The former gives the transparency to characterize the laminar flow over the HOPG, whereas the latter provides enhanced chemical resistance. These robust microfluidic devices are reusable and do not suffer from swelling in harsh solvents. It should be noted that most of the devices used in literature to generate laminar flows and RD conditions are made from PDMS, which drastically limits the solvent choices due to the poor chemical compatibility of PDMS.

Finally, our contribution does not only present the investigation of the molecular diffusion of species and mass transport under laminar flow conditions but also the reaction kinetics including the production of the aryl radical; the consumption of aryl radical by the surface attachment; the formation of iodine; and the deactivation of the aryl radical with iodine. The effects of all these above-mentioned chemical and physical processes which govern the final morphology of grafted layer were systematically demonstrated with numerous experiments and supported by detailed numerical simulations.

3. The microfluidic method for controlling the chemical gradients has been reported in many reports.

We are somewhat disappointed that the Reviewer #2 is trying to weaken the novelty of our work with only general comments, rather than presenting specific arguments or suggestions that might help us to further improve our manuscript. We must say that we are quite aware of the existing literature on controlling chemical gradients via microfluidic approaches. In fact, this topic has been one of the main research areas of our group over the last 10 years. Previously, we have already showed that controlled chemical gradients can lead to: graded metal-organic-frameworks (MOF) thin films (*Adv. Mater. Tech*, 2019, 4, 6, 1800666); films with compositional gradients for photovoltaic applications (*Adv. Eng. Mater*, 2020, 10, 33, 2001308); and/or to the fabrication of reusable SERS substrates for in-situ detection of multiple analytes (*Adv. Science*, 2020, 7, 12, 1903172). Moreover, we have demonstrated that controlled chemical gradients (and RD areas) can also lead to unveiling pathway complexity in a crystallisation process (*Angew. Chem.*, 2021, 13, 29, 16056-16063); controlling defect engineering in crystals (*Applied Mater. Today*, 2020, 20, 100632); and inducing chiral symmetry breaking processes in achiral molecules (*Nat. Commun.*, 2022, 13, 1766). However, in the present manuscript, we go one step further and demonstrate that while the no-slip boundary condition prevailing at the walls of continuous-flow microfluidic devices is often associated with precipitation events that may limit the device's long-term operation and performance, it can also be used to enable a spatiotemporal functionalization of HOPG with 2D and 3D control.

4. The background physics and mechanism for the generation of HOPG have not been deeply studied.

We do not fully understand the comment formulated by the reviewer. The highly oriented pyrolytic graphite (HOPG) used in the experiments presented is a commercial product, and hence, it was purchased (HOPG, grade ZYB, Advanced Ceramics Inc., Cleveland, USA). Therefore, we do not consider it relevant to discuss its fabrication.

If the referee asks about the mechanism of the reaction of the aryl radicals onto the HOPG, we would like to refer to our discussion in the **Supplementary Information 3.2**.

5. The rationality of the numerical simulation method and the reliability of the simulation results are not proved.

We have now included in the revised manuscript 2 new additional sections (**Supplementary Information 3.5-3.6**) describing the hydrodynamic flow conditions for flow rates of 10 to 600 $\mu\text{L min}^{-1}$ (**Fig. S6-S9**), and the parametric analysis conducted with 28 different combinations of k_1 and k_2 (**Supplementary Table 1, Fig. S10-13**). The latter was conducted to estimate the concentration of aryl radicals along the microfluidic device, and to identify the values of k_1 and k_2 that better fit the experimentally-observed layer thicknesses. It should be noted that the results obtained from the simulations performed with the identified k_1 and k_2 pair ($k_1 = 10^{-3} \text{ m}\cdot\text{s}^{-1}$ and $k_2 = 2 \times 10^5 \text{ M}^{-1}\cdot\text{s}^{-1}$) perfectly match the experimental data, not only for a single flow rate (50 $\mu\text{L min}^{-1}$, see **Fig. 2-3**), but also for all the other investigated flow rates (10, 100, 200, 600 $\mu\text{L min}^{-1}$, see **Fig. S23**). Finally, **Fig. S3-S4** clearly demonstrate the validity of the predictions obtained in our numerical simulations, as they are fully consistent with the predictions obtained with previously reported analytical solutions and with numerical works by different authors.

6. The innovation of this study should be clearly explained.

In the revised version of our manuscript, we have modified the *Introduction* to clearly emphasize the novelty of the results presented. Additionally, we have added new sections in the supplementary information to support the experimental data obtained. See for example **Supplementary Information 3.5-3.6** in the revised supplementary information and the new supplementary figures (see **Fig. S6-13 Fig. S16-17**). We have also improved the *Results and Discussion* section by adding a new paragraph that highlights the importance of the controlled gradient formation onto HOPG as well as its prospective applications (please see our answer Reviewer #3's 5th comment). We hope that all these additional data and explanations will clearly show the innovation of our study.

Reviewer #3:

The authors proposed one method to well control the functionalization of graphenic materials by continuous-flow microfluidic devices. This is a detailed report on the device design and the control experiment for the grafting density, layer thickness and morphology of the functionalized graphenic materials, which can be tuned by flow time and flow rate. By combining the numerical simulation with experiments, the manuscript presents the ability to control the grafting process of graphenic materials and tried to figure out the physical mechanism behind the phenomena. This is an interesting paper that will attract some attention from the researchers working on the functionalization technology, while the following specific issues should be improved:

We thank Reviewer #3 for highlighting the importance of our approach and for considering it an interesting method for surface functionalization. We also acknowledge that the reviewer appreciates our efforts to prepare a detailed study and to explain the mechanisms behind the controlled grafting process presented.

1. *The authors highlighted that the better controlling during the microfluidic grafting in the proposed microfluidic devices is thanks to the no-slip boundary condition, both in the introduction and also in the Results. However, they just showed an indirect evident that the*

concentration of intermediate product aryl radical is larger at the top and bottom walls by numerical simulation to support this proposal. Can the author present more direct experimental evidence for this claim? If the no-slip boundary condition doesn't work, what would the morphology of grafted HOPG be like? Is it possible to present the experimental results of this control experiment? In addition, what is the limited length for the no-slip boundary condition to work?.

To provide more detailed information on the crucial role of the no-slip condition prevailing at the device walls, we have now added a new section to the manuscript (**Supplementary Information 3.5**) and 6 new figures (**Figs. S6-9** and **S16-17**). The figures (**Fig. S7-9**) and the new section describe the velocity profiles obtained along the device width (x direction) and device height (z direction), as a function of the imposed flow rates, highlighting the impact of the no-slip condition prevailing at the walls, on the velocity profiles developing inside the microfluidic device. This data clearly indicates the influence of the no-slip condition over the velocities and residence times (i.e., the time during which the reactants can diffuse and react with the HOPG substrate) of the different fluid elements (reactants) flowing through the device. The velocity profiles and, thus, the residence times, vary so much along the height of the device, which is very important in the generation of chemical gradients with 2D and 3D control, and consequently, in the controlled functionalization of the HOPG. Indeed, the fluid elements (and reactant molecules) flowing near the HOPG substrate will (i) have much more time to diffuse along the width of the device (x direction), thus widening the RD region (i.e., the portion of substrate surface where grafting can occur), and (ii) have much more time to undergo the series of reactions associated to the production, deactivation and grafting of the aryl radical to the HOPG substrate.

When the chemical grafting is performed in bulk (i.e., in the absence of stable and controlled velocity and concentration profiles near the liquid-solid interface) the functionalization and morphology of the grafted layer is irregular and nonuniform (see **Fig. S20a**). Moreover, in bulk experiments it is not possible to generate chemical gradients with 2D and 3D control, and consequently, only irregular and nonuniform grafted layers can be obtained. Therefore, it is the stable and controlled velocity and concentration profiles near the liquid-solid interface happening in our devices (operated in laminar flow regime) that allow us to control in 2D and 3D the functionalization. Note that even though in bulk the no-slip boundary condition is present, the chaotic nature of the imposed turbulent flows prevents the development of stable and controlled velocity and concentration profiles near the reactive substrate. Furthermore, because bulk experiments are done in batch (as opposed to the continuous method of our microfluidic experiments) one cannot control the concentration of aryl molecules over the substrate in time and space, hence, one cannot obtain grafted layers with chemical gradients.

Unfortunately, it is not feasible to avoid the no-slip boundary condition experimentally. Nonetheless, in order to better demonstrate the effect of no-slip boundary condition and gain a comprehensive understanding, we have performed a new numerical simulation assuming a hypothetical case in which a free-slip boundary condition is assumed to prevail at the bottom wall of the channel (i.e., HOPG substrate). A free-slip boundary conditions implies that the fluid in direct contact with the substrate can have a velocity that is not null, in opposition to what happens in reality, where the fluid in contact with the substrate is motionless and does not "slip" relative to the substrate (i.e. there is a no-slip boundary condition at the substrate and

any other wall). As seen in **Fig. S16**, because the fluid on the HOPG substrate ($z=0\ \mu\text{m}$, in Fig. S17b) now has a velocity that is not null, its residence time inside the microfluidic device is smaller than in the case with a no-slip boundary condition. Because of that, in the free-slip hypothetical scenario, there would be less time for the diffusion of reagent (and for the reaction to occur) inside the microfluidic channel. As a consequence, the region that could be functionalized would be much smaller (**Fig. S16f-g**, compared to **Fig 2f-g**). Also, the associated concentration profiles, which dictate the characteristics of the resulting grafted layer, would be different both along the length of the microfluidic channel and along its width, if there were free-slip at the HOPG surface (**Fig. S16d,f** compared to **Fig 2d,f**)

We also plotted the velocity profiles at 1 micron from the HOPG along the width (x), and along the height (z) of the microfluidic device, considering free-slip and no-slip boundary conditions (see **Fig. S17**). As expected, at 1 micron from the substrate, the velocity along the width is much higher if there is slippage (1-2 orders of magnitude higher; Fig. S17a). Additionally, the velocity profile along the height highlights how different the flow would be if free-slip would be considered: the maximum velocity would occur at the HOPG instead of at the middle height of the device (Fig. S17b). These differences in velocity (no-slip vs. free-slip) are behind the changes in the concentration profiles that are seen when we compare **Fig S16** and **Fig 2**.

All these new simulation results further prove that the no-slip condition is the key to enlarge the grafting area and to boost the controlled covalent functionalization of HOPG.

2. From the numerical simulation, the RD area can cover at least $300\mu\text{m}$ in zone 2. In experiments, what is the size of the effective RD area (well-controlled morphology of grafted HOPG) at zone 2? From Fig. 3a, the morphology is not so homogenous along the x-axis. The size data of effective RD area would be what the readers care about and useful to evaluate the significance of the proposed devices.

We thank the reviewer for his/her comment that will help us to clarify different aspects of the results presented. As it is shown in **Fig. 3b** (black dots combined with black dashed line), we have experimentally demonstrated that the grafted area covers at least $300\ \mu\text{m}$ ($\pm 150\ \mu\text{m}$ vicinity of the interface) at zone 2 for the experiment performed at a flow rate of $50\ \mu\text{L min}^{-1}$ (for 5 min). This experimental result agrees well with the numerical simulation data (e.g., Fig. 2g). Moreover, **Fig. S21** and **Fig. S22** further prove the covalent functionalization at the corresponding location by employing both Raman spectroscopy and AFM measurements, respectively.

The morphologies observed in the grafted layers mainly depend on the local chemical concentrations of aryl radicals. It should be noted that in our approach we have a different concentration of aryl radicals (*i*) along the width of the main microfluidic channel (x direction), due to the diffusion of reagents across the co-flowing streams and (*ii*) along the length of the main channel (y direction), due to the interplay between continuous formation and deactivation of aryl radicals. Accordingly, it is expected that the morphology will not only change along the x direction but also along the y direction

3. Figure 4a presents the experimental results at a fixed position with varying time. I am confused why the morphology (stripes) of HOPG changes in the four pictures a lot.

We understand the reviewer's concern about the time dependent experiments. We have performed time dependent experiments in separate sets of measurements. Therefore, the experiments for each time point (1 min, 5min, 10min, 60 min) are performed on different HOPG substrates and characterized at the very end of the experiment. For this reason, it is expected that the terraces of the HOPG will be different between consecutive experiments. Note that it would not be feasible to clamp the HOPG substrate such that the microfluidic channel is placed exactly at the same position after the characterization of a sample prepared in 1 minute. Instead, it is preferred to perform the experiments for each data point separately and independently characterize the reactive substrates prepared with different experimental times. The latter approach can avoid any possible experimental error that could occur because of misalignment of the substrate between different time points. As we understand that this may lead to confusion, we have now clarified this concern raised by the reviewer in the figure caption of **Fig. 4**.

4. How does the author define the basal height of the HOPG to obtain the layer thickness of the grafted layer?

Based on the clear topographical information recorded in the AFM image, the line profiles were used to identify the layer thickness based on the corral (empty) area. In order to identify the bottom of the empty corral as the HOPG surface, AFM tip scratching experiments were performed by using the contact scanning mode of the AFM with a force of about 80 nN to locally remove the functionalized layer and expose the bare HOPG surface. The depth of the natural empty corrals has been demonstrated to be in line with that of the scratched empty area, as shown in **Fig. S19**. As shown previously, the AFM tip scratching with a force of about 80 nN has no morphological impact on the bare HOPG surface (*J. Phys. Chem. C* 2021, 125, 21624–21634).

This has been further clarified in **Supplementary Information 4.2** of the revised Supplementary Information: "*Line profiles were used to identify the layer thickness based on the corral (empty) area (see the example in **Supplementary Fig. 19**). In order to identify the bottom of the empty corral as the HOPG surface, AFM tip scratching experiments were performed by using the contact scanning mode of AFM with a force of about 80 nN to locally remove the functionalized layer and expose the bare HOPG surface. The AFM tip scratching with a force of about 80 nN has been shown to have no morphological impact on the bare HOPG surface (*J. Phys. Chem. C* 2021, 125, 21624–21634).*" The new reference has been added in the Reference section of the Supplementary Information: "*J. Phys. Chem. C* 2021, 125, 21624–21634".

5. The authors have changed parameters, such as the flow time, flow rates, to see how the grafting process occur on the HOPG by identifying the thickness of grafted layer, size and density of corrals. Upon changing these parameters, the morphology of the grafted HOPG changes a lot. I suggest the author to summarize the optimal condition to obtain the monolayer and bilayer homogenous grafted layer on the HOPG at the end of the Results and discussions. Additionally, how do the thickness of grafted layer, size and density of corrals of HOPG affect its functionality?

We thank the reviewer for his/her recommendation. As requested, at the end of the *Results and Discussions* section of the revised manuscript, we have given a summary regarding the

control of the aryl grafted layer simply by taking advantage of the flow conditions: “The results described above show that our microfluidic approach offers the possibility to generate chemical gradients onto HOPG with 2D and 3D control. As shown in **Supplementary Table 2**, we have demonstrated that at medium flow rates ($50 \mu\text{L min}^{-1}$), multilayer grafted layers of aryl moieties can be generated and tailored from 1.5 nm to 3.0 nm in thickness by adjusting the flow time from 1 minute to 120 minutes. Moreover, we have also observed that lower ($10 \mu\text{L min}^{-1}$) and higher ($600 \mu\text{L min}^{-1}$) flow rates induce the formation of a monolayer (1 nm). This controlled functionalization of a graphenic surface is highly relevant in the surface science field as it can lead to new functions and applications. For example, the layer thickness, size and density of corrals can strongly influence the on-surface properties of HOPG, such as its hydrophilicity. Additionally, the spatially controlled grafting can also be explored to fine-tune the interaction with external adsorbates, and this can be extremely relevant in the context of developing new sensing technologies (Angew. Chem. Int. Ed. 2022, 61, e202200115). The spatially-controlled grafting of aryl moieties onto HOPG can also provide unique experimental conditions and opportunities, e.g. to unveil the impact of nanoconfinement on the formation of self-assembled molecular networks in one go by employing one of our microfluidic grafted HOPG substrates (ACS Nano 2016, 10, 10706–10715). Note that a variety of different nanoconfinement conditions can be generated in a single substrate via the controlled gradient formation prevailing inside our microfluidic device”.

Additionally, the following table has been added in the *Supplementary Information*:

Table S2. The grafting layer thickness, size and density of corrals in the centre of the zone 2 as the function of the flow time (min) and flow rate ($\mu\text{L}/\text{min}$).

Flow rate	10 $\mu\text{L}/\text{min}$			50 $\mu\text{L}/\text{min}$			100 $\mu\text{L}/\text{min}$			200 $\mu\text{L}/\text{min}$			600 $\mu\text{L}/\text{min}$		
Flow time	a	b	c	a	b	c	a	b	c	a	b	c	a	b	c
1 min				1.5	60.8	30.3	1.3	72.6	7.7	1.2	82.5	12.3			
5 min				2.0	55.2	29.0									
10 min	1.0	61.9	13.7	2.1	49.7	24.7							0.9	42.8	53.0
60 min	2.0			2.2	41.3	31.7									
120 min				3.0	46.3	9.3									

a: average layer thickness (nm); **b:** average diameter of corrals (nm); **c:** average number of corrals per square micrometers.

6. In this manuscript, the author compared the morphology between the proposed microfluidic devices and those obtained from bulk method. Does the functionalized grahenic materials obtained in this manuscript work as well as those obtained from the bulk method, or much better? Can the authors add some control experiments?

The differences between the outcome of the bulk drop-casting method and the microfluidics method in terms of morphology are discussed in the manuscript (see the discussions of **Fig. 3** and **Fig. S20** in the main text). We also refer to a control experiment (see the discussion of **Fig. S20** in the main text) whereupon premixing of the reagents, the microfluidics experiment gives the same result as what happens upon bulk drop-casting. These experiments highlight that the exquisite control over concentration gradients established inside our microfluidic device can be leveraged to enable a controlled spatiotemporal functionalization of millimetre-size reactive surfaces in 2D and 3D. To evaluate the universality of our microfluidic approach, we did additional experiments regarding the modification of gold substrates with 1-dodecanethiol using our microfluidic device, obtaining interesting results (see the reply to

Reviewer #1's 2nd comment and **Fig. R1**) that also demonstrate the controlled transfer of chemical gradients onto a different substrate.

Please refer to our reply to 5th comment concerning the functionality. For instance, the big difference in morphology between the modified surfaces created by the microfluidics approach (where a lateral gradient of nanocorrals density and diameter, as well as the layer thickness, are precisely controlled) compared to the drop casting approach (where nanocorrals' size and shape cannot be controlled), leads to unique experimental conditions and opportunities, that will be studied in our group in the near future. For example, one could unveil the impact of nanoconfinement on the formation of self-assembled molecular networks in one go by employing one of our microfluidic grafted HOPG substrates. Note that a variety of different nanoconfinement conditions can be generated in a single substrate via the controlled gradient formation prevailing inside our microfluidic device.

Reviewer comments, further round review –

Reviewer #1 (Remarks to the Author):

The paper has been thoroughly corrected by the authors, they have answered all my questions. The paper can now be published

Reviewer #2 (Remarks to the Author):

The authors have modified this manuscript according to the comments. I would like to recommend accepting it as this version.

Reviewer #3 (Remarks to the Author):

My comments had been addressed point by point and the corresponding discussions in manuscript has been revised. I have no further comments on this manuscript now.